# Multi-photon near-infrared emission saturation nanoscopy using upconversion nanoparticles

Chaohao Chen [1], Fan Wang [1], Shihui Wen [1], Qian Peter Su[1], Mike C.L. Wu[2], Yongtao Liu[1], Baoming Wang [1], Du Li[1], Xuchen Shan[1], Mehran Kianinia[1], Igor Aharonovich[1], Milos Toth [1], Shaun P. Jackson[2], Peng Xi[3] & Dayong Jin [1,4]

Multiphoton fluorescence microscopy (MPM), using near infrared excitation light, provides increased penetration depth, decreased detection background, and reduced phototoxicity. Using stimulated emission depletion (STED) approach, MPM can bypass the diffraction limitation, but it requires both spatial alignment and temporal synchronization of high power (femtosecond) lasers, which is limited by the inefficiency of the probes. Here, we report that upconversion nanoparticles (UCNPs) can unlock a new mode of near-infrared emission saturation (NIRES) nanoscopy for deep tissue super-resolution imaging with excitation intensity several orders of magnitude lower than that required by conventional MPM dyes. Using a doughnut beam excitation from a 980 nm diode laser and detecting at 800 nm, we achieve a resolution of sub 50 nm, 1/20th of the excitation wavelength, in imaging of single UCNP through 93 μm thick liver tissue. This method offers a simple solution for deep tissue super resolution imaging and single molecule tracking.

[1] Institute for Biomedical Materials and Devices (IBMD), Faculty of Science, University of Technology Sydney, Sydney, NSW 2007, Australia. [2] Heart Research Institute, and Charles Perkins Centre, University of Sydney, Camperdown, NSW 2006, Australia. [3] Department of Biomedical Engineering, College of Engineering, Peking University, 100871 Beijing, China. [4] ARC Research Hub for Integrated Device for End-user Analysis at Low-levels (IDEAL), Faculty of Science, University of Technology Sydney, Sydney, NSW 2007, Australia. These authors contributed equally: Chaohao Chen, Fan Wang. Correspondence and requests for materials should be addressed to F.W. (email: fan.wang@uts.edu.au) or to D.J. (email: dayong.jin@uts.edu.au)

Multiphoton fluorescence microscopy (MPM)[1], also known as nonlinear, two-photon or three-photon scanning microscopy, uses near-infrared (NIR) excitation that penetrates as deep as hundreds of microns[2] through the so-called transparent biological window[3]. NIR light does not excite autofluorescence background and has minimal phototoxicity, ideal for live-cell imaging. However the longer wavelength used in MPM leads to lower resolution, hindered by the diffraction limit ($\lambda/2NA$)[1,4]. Stimulated emission depletion (STED) approach can minimize the area of illumination at the focal point to achieve super resolution imaging[5–7]. This approach has enabled MPM[8,9] to deliver five folds improvement in resolution in imaging fixed cells. Employing two spatially aligned and temporally synchronized femtosecond lasers in a sophisticated pulsed STED[10,11], a super resolution of 60 nm at a depth of 50 μm has been achieved for imaging dendritic spines in living brain tissue[12,13]. Nevertheless, it requires high-power and expensive femtosecond lasers for MPM to achieve super resolution, due to low efficiency of the nonlinear multiphoton process, e.g., small multiphoton absorption cross-sections of the probes[9].

Upconversion nanoparticles (UCNPs) are typically doped with ytterbium sensitizer ions ($Yb^{3+}$), which absorb the photons located at NIR region and transfer their excitation to activator ions[14,15]. Thulium ($Tm^{3+}$) co-doped UCNPs, capable of converting 980 nm NIR photons into strong 800 nm NIR emissions, are promising for background-free deep tissue imaging. The sensitizer $Yb^{3+}$ ions have much larger absorption cross-section in NIR than that of dye molecules used in MPM[16,17], benefiting from its linear absorption. Moreover, the photon upconversion process is a sequential absorption process where the activator ions, such as $Tm^{3+}$, have multiple long-lived intermediate states to facilitate multiphoton upconversion process[18]. Therefore, UCNPs bypass the crucial requirement of simultaneous absorption of more than two photons from the femtosecond lasers, providing several orders of magnitude lower excitation threshold than that for most efficient multiphoton dyes[17].

UCNPs have been recently discovered suitable for STED-like super resolution nanoscopy with sub-30 nm optical resolution in resolving the cluster of single UCNPs[19], and a resolution of 80 nm has been further demonstrated in resolving subcellular protein structures[20]. However, this scheme is not suitable for deep tissue super resolution imaging, because the 4-photon upconversion emissions, at wavelength around 455 nm, will attenuate rapidly through deep tissue. Ideally a NIR-in and NIR-out configuration, by detecting emissions above 700 nm from UCNPs, would enable upconversion nanoscopy imaging in deep tissue.

Here we report that setting the 980 nm excitation laser to create a doughnut beam enable super resolution imaging of single UCNPs through deep tissue. Low power coherent excitation at 980 nm can easily saturate the metastable level that emits bright NIR emission (800 nm), and the nonlinear power-dependent emission curve (saturation curve) is sharp. Both play the essential role in enabling a new mode of near-infrared emission saturation (NIRES) nanoscopy, ideal for deep tissue super-resolution imaging, with remarkable light penetration depth, ultra-low auto-fluorescence background and minimum photo-toxicity. We further demonstrate that fine tuning of the doping concentration and material design of UCNPs can further improve their optical properties towards enhanced resolution, suggesting a new scope for material science community to investigate more efficient super resolution probes through deep tissue.

## Results

**Advantage of UCNPs in deep tissue imaging**. To meet the requirement of super-resolution imaging through deep tissue, we

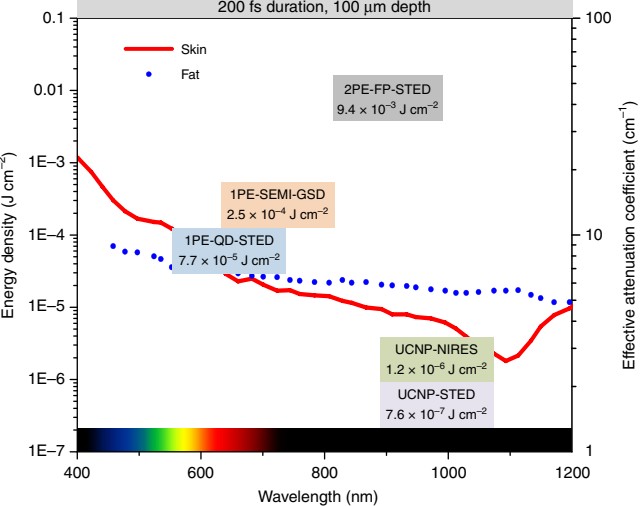

**Fig. 1** The energy densities required by various probes for deep tissue super-resolution imaging. In order to facilitate the comparison, we normalize the excitation/depletion power to certain pulse period (200 fs) through 100 μm tissue[11,19,21,22]. The red and blue curves show the light effective attenuation coefficient in the tissue, adapted by permission from Springer Nature ref. [45] Copyright (2009). 1PE, one-photon excitation; 2PE, two-photon excitation; QD, quantum dots; FP, fluorescence protein; SEMI, semiconductor nanowires segments

first calculate and examine the minimum excitation/depletion energy density (J cm$^{-2}$) for fluorescent proteins[11], quantum dots[21], semiconductor nanowires segments[22], and UCNPs[19] used in STED and ground state depletion (GSD) approaches (Fig. 1). The normalized calculation of energy densities for deep tissue imaging is shown in Supplementary Note 1 and key parameters of these probes are summarized in Supplementary Table 1. The tissue attenuates more power for shorter wavelength, which requires large power in visible range to achieve high resolution, if using one-photon excitation (1PE) scheme. That is why two-photon excitation (2PE) in NIR range is the commonly used method. But multiphoton probes have extremely small absorption cross-sections, thereby requiring even higher excitation power. In contrast, taking both advantages of NIR excitation wavelength and large absorption cross-section, UCNPs provide a potential solution by only requiring small laser power through the deep tissue.

**Emission saturation enabled sub-diffraction resolution**. NIRES nanoscopy takes the advantage of multi intermediate ladder-like energy levels of UCNPs (Fig. 2a), easily converting 980 nm photons into 800 nm photons. The emission saturation curve of the two-photon state $^3H_4$ (800 nm) (Fig. 2b) shows an early onset of upconversion emissions at low excitation power density and sharp rising-up slope, reflecting its non-linear energy transfer assisted photon upconversion process. The decrease in the saturation curve under high excitation power (larger than 2 mW) is due to the energy being further up-converted from $^3H_4$ to the higher energy states. To generate a super-resolution image of single nanoparticle by NIRES, a tightly-focused and doughnut-shaped excitation beam is used to scan across a sample containing UCNPs. Only when a single UCNP sits in the middle of the doughnut beam, NIRES generates a negative contrast. By using the definition of resolution in GSD microscopy[23], a full-width at half-maximum (FWHM) of the dip at the measured point spread function (PSF) of a single UCNP defines the experimental resolution achieved by NIRES (Fig. 2c). By taking the advantage of the

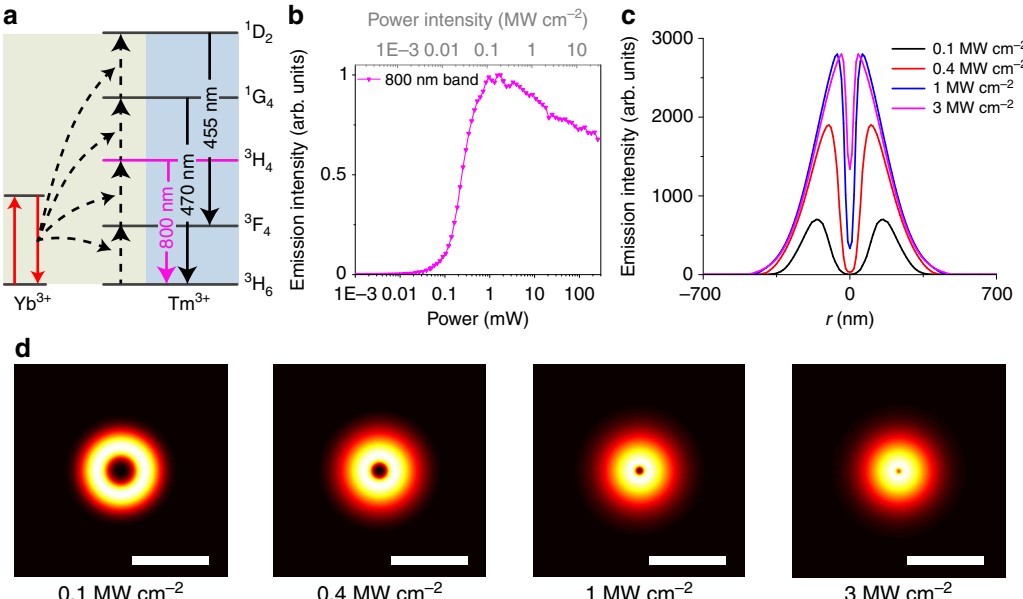

**Fig. 2** The principle of NIRES nanoscopy using UCNP as multiphoton probe for deep tissue imaging. **a** The simplified energy levels and upconversion process of $Yb^{3+}$ and $Tm^{3+}$ co-doped UCNPs. The sensitizer $Yb^{3+}$ ions initiate the photon upconversion process by a linear and sequential absorption of 980 nm excitation. Due to the multiple long-lived intermediate states, the energy is stepwise transferred onto the scaffold energy levels of emitters $Tm^{3+}$, eventually facilitate multiphoton upconversion emission, including emissions from the four photon upconversion excited state $^1D_2$ (455 nm), three photon state $^1G_4$ (470 nm), and two photon excited state $^3H_4$ (800 nm). **b** The saturation intensity curve of the 800 nm emissions from UCNPs (40 nm NaYF$_4$: 20% $Yb^{3+}$, 4% $Tm^{3+}$) under 980 nm excitation. **c** Cross-section profiles of the saturated upconversion emission of UCNPs at four different excitation powers of 0.1, 0.4, 1, and 3 MW cm$^{-2}$. **d** The simulated "negative" contrast images of the cross-section profiles of a single UCNP. Pixel size, 10 nm. Scale bar is 500 nm

uniform size of monodisperse UCNPs, the actual resolution of NIRES can be calculated through deconvolution with the size of a single nanoparticle (see Supplementary Note 2, the fitting parameters are summarized in Supplementary Table 2). With the nonlinear excitation process, the optical resolution of NIRES is limited by diffraction limitation under low excitation power (Fig. 2d, 0.1 and 0.4 MW cm$^{-2}$) but this limitation can be easily broken with high power (Fig. 2d, 1 and 3 MW cm$^{-2}$) using a purpose-built NIRES setup, shown in Supplementary Note 3 and Supplementary Fig. 3.

The resolution of NIRES at a certain excitation power density is essentially determined by the emission saturation curve (Fig. 2b). In our case, there are three features from the curve affecting the resolution. The first feature is the power point ($I_S$) to achieve the half value of the maximum emission intensity. The second feature is the power point ($I_{MAX}$) to achieve the maximum emission intensity with fixed $I_S$, in other words smaller $I_{MAX}$ indicates more superlinear shape of the curvature between $I_S$ and $I_{MAX}$. The third feature is the onset value of the curve, which is defined as the power point (in unit of $I_S$) to achieve e$^{-2}$ of the maximum emission intensity. Larger onset indicates more underlinear shape of the curvature between 0 and $I_S$. Lower values of $I_S$ and/or $I_{MAX}$ decrease the size of dark spot in the emission donut pattern, measured by the FWHM value of the dip, thereby enhancing the resolution. Larger onset of the curve offers lower depth of the donut emission PSF to yield better resolution.

**Activator doping enhanced resolution**. The saturation curve of UCNP can be optimized by tuning the doping concentration of emitters (Supplementary Fig. 6). UCNPs with lower $Tm^{3+}$ doping concentration can be easily saturated with lower values of $I_S$ and $I_{MAX}$ (Supplementary Fig. 6) due to their smaller energy transfer ratio[20] and resultant lower saturated carrier flow rate, which is proportional to laser induced carrier generation rate. The

lower values of $I_S$ and $I_{MAX}$ are favorable towards achieving higher resolution. However, lower $Tm^{3+}$ doped UCNPs have smaller onset of their saturation curve (2% $Tm^{3+}$ in Supplementary Fig. 6c), which substantially affects the resolution. Therefore, it is hard for UCNPs with 2 and 3% $Tm^{3+}$ to achieve higher resolution even by increasing excitation power (Fig. 3). UCNPs with high $Tm^{3+}$ doping concentration have shaper curvature of their onset of the saturation curves, but the values of $I_S$ and $I_{MAX}$ (Supplementary Fig. 6b) are too high because of larger energy transfer ratio[20] and higher cross-relaxation rate[19] for high $Tm^{3+}$ doped UCNP, therefore do not benefit to achieve higher resolution. As shown in Fig. 3, UCNPs with 6 and 8% $Tm^{3+}$ require higher power to achieve the same resolution as that for 4% $Tm^{3+}$ UCNPs, optimized for NIRES (Supplementary Fig. 1c). With that, the highest resolution for singe nanoparticle imaging is 33.9 ± 12.3 nm (Fig. 3 insert) at an excitation power density of 4 MW cm$^{-2}$. This resolution can be further improved by optimizing the sensitizer concentration or designing a core-shell structure, suggesting a large scope for materials science community to improve the resolution of NIRES nanoscopy. Note that the conventional method of square root law cannot be used to fit the resolution of NIRES due to the unique saturation curve of UCNPs. Supplementary Note 1 provides more details on the fitting method to calculate the theoretical resolution in imaging single UCNP by NIRES (Supplementary Fig. 2). Supplementary Note 5 and Supplementary Fig. 7 provide more data using NIRES to resolve single UCNPs inside HeLa cell with a resolution of 65 nm in biological environment.

**Deep tissue imaging by NIRES**. We further examine the penetration depth and resolution of NIRES imaging through deep tissue (Fig. 4). In this experiment, 4% $Tm^{3+}$ 40% $Yb^{3+}$ co-doped UCNPs are attached behind a 93 μm thick slice of mouse liver tissue (Supplementary Note 6 and Supplementary Fig. 8), which

allows UCNPs diffuse into the tissue slice for super-resolution imaging of single UCNPs from different depth (Fig. 4a). Due to the aforementioned strong attenuation for visible emissions

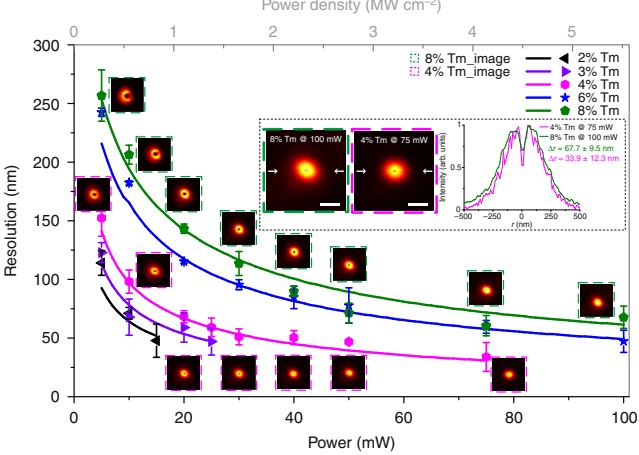

**Fig. 3** Super-resolution scaling $\Delta r$ of UCNPs as a function of the excitation power (intensity). The dots of experimental data are fitted well to the simulation results (solid lines; Supplementary Fig. 5). Error bars are defined as s.d. from line profiles of several measurements. Insets, NIRES images of 8% $Tm^{3+}$ doped UCNPs at 5.5 MW cm$^{-2}$ (left), 4% $Tm^{3+}$ one at 4 MW cm$^{-2}$ (middle), and the corresponding cross section profile lines (right). Pixel dwell time, 3 ms; pixel size, 10 nm. Scale bar is 500 nm. The series of NaYF$_4$: 20% $Yb^{3+}$, $x$% $Tm^{3+}$ UCNPs ($x$ = 2, 4, 6, and 8) ~40 nm in diameter are controlled synthesized and characterized by following previously reported methods[19,46], and shown in Supplementary Note 4 and Supplementary Fig. 4

(shown in Fig. 4b and Supplementary Fig. 9), through a 93 μm liver tissue slice, there is only 11.3% of 455 nm emission left in confocal imaging (Fig. 4c), and in contrast there are 38.7% of strong signal at 800 nm detectable in both confocal imaging (in red) and NIRES (in magenta) super resolution imaging. More encouragingly, by increasing excitation power to compensate the aberration induced distortion on excitation PSF[24], a fairly consistent resolution of sub-50 nm has been maintained without any aberration correction through a tissue as deep as 93 μm (Fig. 4d). The lower refractive index[25,26] of tissue for NIR light results in less aberration than that for visible beam, which also contributes to the high resolution achieved by NIRES for single nanoparticle imaging.

Figure 5 further examines the resolution power of NIRES in resolving single UCNPs from nanoparticle clusters in deep tissue. Figure 5a shows a bright field image of an 88 μm liver tissue slice, merged with the fluorescence image of four clusters of UCNPs (in magenta). At this depth, 800 nm confocal microscopy (Fig. 5b) does not provide sufficient resolution to image single UCNPs within a diffraction limit area (Fig. 5c, d). In contrast, NIRES nanoscopy can clearly resolve single UCNPs by either negative (Fig. 5e, f) or positive images (Fig. 5g, h). The two typical areas of confocal and NIRES/NIRES+ images, marked by orange square (Fig. 5c, e, g) and green square (Fig. 5d, f, h) respectively, clearly illustrate the sub-50 nm resolution achieved by NIRES nanoscopy, in resolving single UCNPs separated with 161 nm and 275 nm (Fig. 5i, j). It is notable that the resolvable distance of 161 nm is not the limit. For instance, UCNPs with a distance 72 nm can be resolved in Supplementary Fig. 12. Supplementary Fig. 13 provides additional data showing that the best resolutions of 54 and 62 nm have been achieved through an 85 μm mouse kidney tissue slice and a 92 μm brain tissue slice, respectively. From the

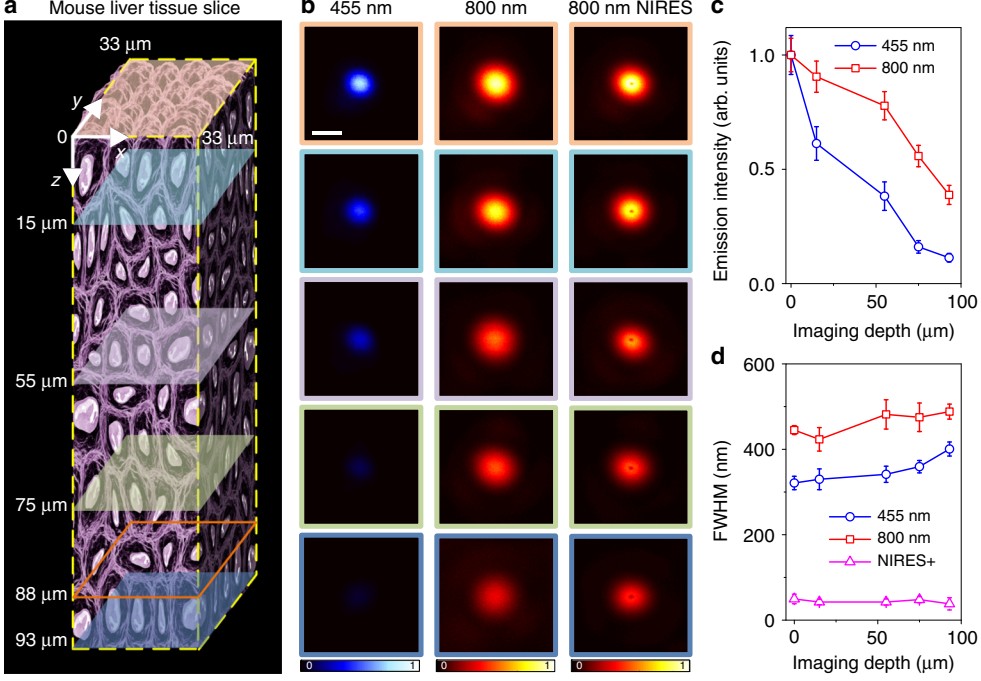

**Fig. 4** The optical resolution of different imaging modalities at different depth of a liver tissue slice. **a** An illustration of a mouse liver tissue slice with 93 μm thickness. **b** Single particle imaging at different depth in liver tissue. Left, confocal images from 455 nm emission; middle, confocal images from 800 nm emission; right, the corresponding NIRES images. **c** The normalized emission attenuation at different depth through the liver tissue. **d** The corresponding FWHM in **b**; The resolutions of NIRES in **d** are 49.6 ± 11.1 nm (0 μm), 42.4 ± 6.2 nm (15 μm), 42.4 ± 7.2 nm (55 μm), 48.0 ± 7.3 nm (75 μm), 38.2 ± 14.3 nm (93 μm). Detailed data in **d** is shown in Supplementary Table 3. Benefiting from the controlled synthesis of intensity monodispersed UCNPs, each single nanoparticle can be distinguished from their clusters from clusters by comparing their emission intensity to statistical averaged value[37,46–49]. Error bars are defined as s.d. Pixel dwell time for confocal and NIRES is 3 ms. The pixel size for confocal and NIRES is 10 nm. The scale bar is 500 nm

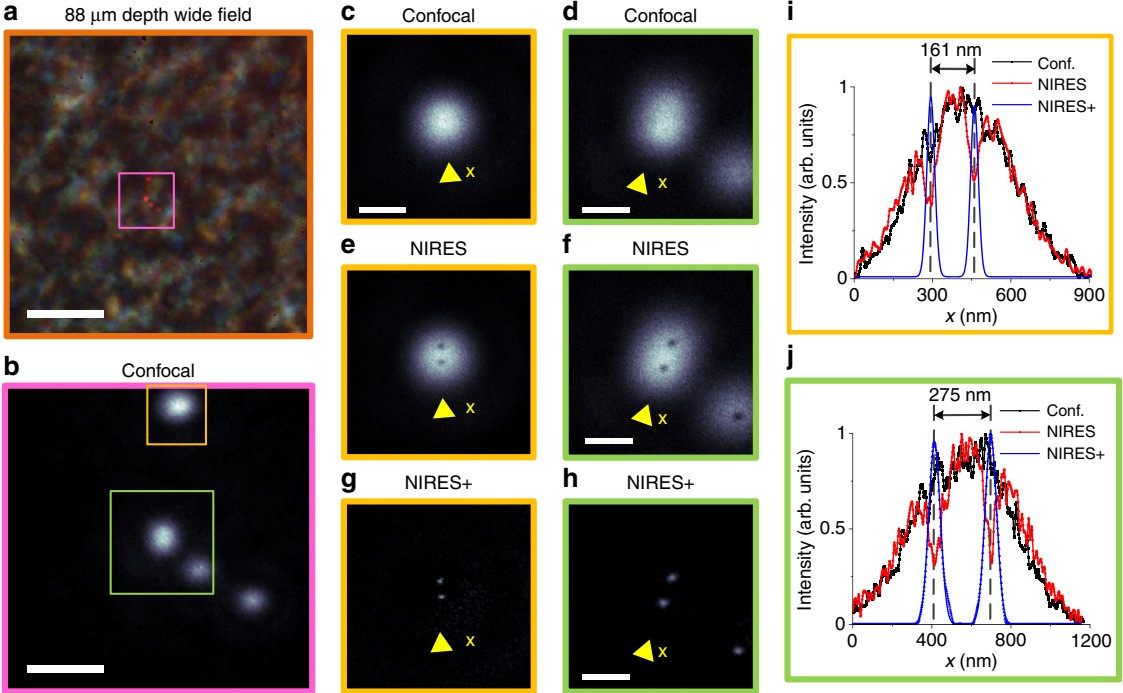

**Fig. 5** NIRES nanoscopy for super resolution imaging of single UCNPs through deep mouse liver tissue. **a** Bright field and fluorescence wide field image of UCNPs at 88 μm depth inside a mouse liver tissue slice. **b** Confocal images of a 6 μm × 6 μm area containing clusters of UCNPs. **c** and **d** are the zoom-in confocal images of two areas of interest from **b**. **e** and **f** are the raw data of NIRES images of **c** and **d**. **g** and **h** are the processed data of NIRES+ images of **e** and **f**. **i** Line profiles of UCNPs from the confocal image (**c**), raw NIRES image (**e**), and positive NIRES image (**g**). **j** Line profiles of UCNPs from confocal image (**d**), raw NIRES image (**f**), and positive NIRES image (**h**). Pixel dwell time for confocal and NIRES is 3 ms. The Pixel size for confocal and NIRES is 10 nm. The scale bar is 8 μm in **a** and 1.5 μm in **b** and 500 nm in **c**–**h**. Note that the excitation power density to achieve the best resolution (see Supplementary Fig. 10 & 11) in the depth of 88 μm is 5.5 MW cm$^{-2}$, to compensate the scattered excitation power through the thick tissue

high brightness (20,000 photon counts per second, in contrast to the low detection background of 1000 photon counts per second) of a single UCNP emitting at 800 nm through a 93 μm tissue, we estimate the limit in imaging depth to detect a single nanoparticle by NIRES should be around 175 μm. An objective lens with longer working distance (around 100 μm used in our current setup) should be used to confirm this limit in future works.

## Discussion

Compared with our recent work reporting low excitation power STED-like super resolution application[19], the NIRES approach presented here achieves the same level of imaging resolution (<50 nm) using a simple setup. NIRES offers a great deal of simplicity and stability, as an advanced tool to achieve super resolution in deep tissue, superior to the conventional MPM approaches[9,27,28]. In a most recent work, parallel to this report, Wu et al. demonstrated a fluorescence emission difference (FED) microscopy using UCNPs[29], with 172 nm resolution achieved using 800 nm 10 MW cm$^{-2}$ excitation and 660 nm emission detection. With the goal to achieve super-resolution imaging using low laser power and longer wavelength, i.e., to reduce photo toxicity applied to live cells, controlled synthesis of UCNPs will further optimize energy transfer process and resultant saturation intensity properties for bio-photonics to surpass the diffraction limit. The recent progress on functionalization of UCNPs, enabling bio-conjugation[30–33], cell optogenetics[34–36] and long-term tracking in cell[37–40], will empower NIRES to track more biological events. The tuning of multiple emission colors[41,42] and lifetimes[43] of UCNPs will allow NIRES for multiplexed super-resolution imaging of subcellular structures and long-term tracking of single molecules in deep tissue[44].

## Methods

Details and any associated references are provided in the Supplementary Information.

**Data availability**. All the relevant data are available from the correspondence authors upon reasonable request.

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

## Acknowledgements

The authors acknowledge the financial support from the Australian Research Council (ARC) Future Fellowship Scheme (D.J., FT 130100517), Discovery Projects Scheme (I.A. and M.T., DP18010007) and the National Natural Science Foundation of China (61729501, 51720105015). C.C., Y. L. and B.W. acknowledge the financial support from China Scholarship Council scholarships (Chaohao Chen: No. 201609750009; Yongtao Liu: No. 201607950010; Baoming Wang: No. 201606170066).

## Author contributions

F.W. and D.J. conceived the project and supervised the research. F.W. and C.C. designed experiments. C.C., Y.L. and X.S. performed the optical experiments. C.C., F.W. and M.K. conducted the optical setup. C.C., F.W., Y.L. and P.X. built the simulation model. S.W. and D.L. synthesized the upconversion nanoparticles. Q.S., M.W. and S.J. prepared the mouse tissue slices. B.W. performed the cell culture. I.A. and M.T. provided the constructive suggestions. C.C., F.W. and D.J. analyzed the results, prepared the figures and wrote the manuscript. All authors participated in editing of the manuscript.

## Additional information

**Competing interests:** The authors declare no competing interests.

