## [Peer Review File · Nature Communications]

Reviewers' comments:

Reviewer #1 (Remarks to the Author):

This is a new report for nanoscopy by near-infrared emission saturation (NIREs) from the same research group where STED nanoscopy based on UCNPs was reported. I would say this new NIREs is a better version of nanoscopy than the previous one based in STED because it needs only one excitation laser at 980 nm: simpler way to do nanoscopy. Manuscript is well written and data looks convincing, so I would recommend this manuscript can be published in its current form.

Reviewer #2 (Remarks to the Author):

This paper entitled "Multi-photon near-infrared emission saturation nanoscopy using upconversion nanoparticles" reports a near-infrared emission saturation (NIREs) nanoscopy for super-resolution using lanthanide-doped upconversion nanoparticles. Setting a 980 nm excitation laser to create a doughnut beam and detecting at 800 nm, the authors achieved a spatial resolution of 34.4 nm, benefiting from the easily saturable upconversion luminescence. Intercellular imaging to track multiple nanoparticles is also demonstrated. Although the results are interesting, the novelty of this work is too weak to fulfill the publication requirements of Nature Communications, referring to my comments below. Thus, I do not recommend it for publication. I suggest the authors submit it to a specific journal.

1. The principle of this work is essentially the same with a few works, such as ESSat microscopy (Yang et al. Nature Photonics 2015, 9, 658), "negative" GSD (Rittweger et al. EPL 2009, 86, 14001) and fluorescence emission difference microscopy (FED) (Kuang et al. Scientific Reports 2013, 3:1441). In these techniques, a doughnut-shaped beam is used for excitation and a full-width at half-maximum of the dip at the central of the pattern determines the resolution of microscopy. Emission saturation of fluorophores is the key feature which is used to increase the imaging resolution (Yang et al. Nature Photonics 2015, 9, 658; Zhao et al. Optics Express 2016, 23596), which is also used in this work. In addition, the reviewer notices that use of upconversion nanoparticles for ESSat or FED microscopy, utilizing their emission saturation property, has been also reported by another group recently (Wu et al. Optics Express 2017, 25, 30885). I do not see any conceptual innovation from the current work to fulfill the publication requirement of Nature Communications.

I also notice that in the ref. Wu et al. Optics Express 2017, 25, 30885 the authors used 800 nm excitation and detecting at 650 nm. The working wavelengths (980 nm excitation, 800 nm emission) used in the current work are not advantageous.

2. The authors compare a series of nanoparticles with different Tm concentrations. I am not a material expert. But to me it is just screening the nanoparticles. It is lack of an in-depth study to construct a connection between nanoparticle engineering and emission saturation properties, thus seeming lack of guiding significance.

3. The size of nanoparticles used in this study is around 40 nm in diameter. Then the argued resolution of 34.4 nm does not make sense. In addition, these nanoparticles as others like QDs are difficult to modify and functionalize, so they are not advantageous compared to organic biomarkers.

4. The dwell time used is 3 ms, which is much longer than those used for organic biomarkers, typically tens of ns. Such long dwell time would limit the imaging speed.

reports an optical approach to turn on and off the UV emission from NaYF₄:Yb,Tm nanoparticles by modulating excitation pulse, specifically varying pulse width or frequency with constant duty cycle. There is no breakthrough or development in the method compared to their own previous work (J. Mater. Chem. 2011, 21, 18530) and other groups' work (Nat. Nanotechnology 2015, 10, 237) on the

control upconversion emission color in other upconversion systems by excitation modulation. In this paper, the authors just move to the NIR and UV bands of NaYF₄:Yb,Tm nanoparticles. Technically, the authors do not provide enough explanation to the observed pulse-width dependence of NIR and UV emission intensities. Thus I cannot recommend it for publication in Nature Communications. I hope the authors can consider my concrete comments below to improve the manuscript in order for it to be published in a more specific journal.

1. The reviewer thinks the observed pulse width dependence of the UV and NIR emission can be explained or even calculated by considering their impulse response function (to 980 nm excitation pulse) characterized with certain rise and decay time constants (Nanoscale 2017, 9, 1676). The reviewer is expecting longer rise time and shorter decay time from the UV band compared to the NIR band. However, in Fig. 2a the UV band shows both shorter rise and decay time than the NIR band, which the reviewer thinks would not lead to the observed emission pulse-width dependence. Please carefully redo luminescence decay measurements.
2. A numerical simulation study to investigate the pulse-width effect would be also interesting to provide explanation to the observed phenomena if (Figs. 4 and 5) analytical approach is absent.
3. Upconversion emission properties are highly excitation power-density dependent. Excitation intensity should be measured and specified for Figs. 4 and 5.
4. Fig. 6 is inconsistent with the text.
5. Supplementary figures are not cited and discussed in the main text.
6. Some other important literatures regarding use of pulse excitation for upconversion nanoparticles should be also cited.

The authors present results on multi-photon near-infrared emission saturation nanoscopy using upconversion nanoparticles. The upconversion nanoparticles used in their paper allow multi-photon excitation with low laser intensity. Moreover, both the excitation and emission light are in the near-infrared wavelength range, potentially facilitating the imaging of thick samples. On top of the multi-photon excitation, a technique similar to ESSat or GSD is applied to break the diffraction limit and to achieve super-resolved imaging.

The approach and the results presented here are very interesting and promise a simple and powerful way of performing super-resolution imaging in deep tissue. Considering the potential impact of this work, I recommend publication. However, there are a few issues that the authors should take into account:

1. It is not clear how the light intensities in Figure 1 are calculated for various imaging modalities. Please add more information.
2. Please describe briefly the diagram in Figure 2a.
3. Line 117: “the dots of experimental data are fitted well to the simulation results”. It is not clear what type of simulation was performed and how the data were fitted. Please provide more details.
4. In ESSat or GSD, the resolution is proportionally to the inverse of the squared root of the depletion intensity; what is the quantitative relationship between the resolution and the laser intensity in NIREs?
5. Line 135: “further increase of excitation power results in a high background of dip The value of center dip of the doughnut distribution is often above zero”. Please give some quantitative values. What is the ratio of the intensity at the doughnut center to the max intensity of the beam? It is important to evaluate this value carefully, especially considering this ratio could be further compromised when imaging through thick samples. Is this ratio constant when increasing the laser power? In Figure S5, it seems that the intensity at the center dip is more pronounced for UCNPs with lower TM concentration. If it is true, please explain it.
6. Line 157: “The low saturation intensity at intermediate excited states, as upconversion emissions at the peripheral areas away from the doughnut center are ready saturated”. It is not clear what message this sentence is trying to deliver. Please rephrase the sentence.

We would like to thank three reviewers and the editorial office for taking their time and writing constructive comments to improve the quality of our work.

We believe our revised manuscript has addressed all the questions and comments from editor and three referees. Our detailed responses to the reviewers' comments (in blue), the actions taken (in bold black) and the text change in paper (in red) below.

Major change:

Major action 1:

To prove the power of NIRES nanoscopy in deep tissue imaging, we have conducted systematic deep tissue super-resolution experiment. The tissue sample includes liver, brain and kidney. **The details of sample preparation of tissue are added as a new Supplementary Section:**

Section VI. Tissue sample preparation.

Mice post euthanasia with an injection of lethal dose of Xylazine and Ketamine mixture, the mice were transcardially perfused with saline to remove blood content. Brain, kidney and liver tissue samples were collected and fixed in 4% PFA overnight at 4 °C, and sectioned into 100 μm, 150 μm, 200 μm in thickness, using an automated vibratome (Leica VT1200 S). The brain, kidney and liver sections were then mounted in glycerol containing 0.05 mg/ml UCNPs for NIRES imaging. All procedures performed on mice were approved by Animal Care and Use Committee, the University of Sydney Animal Ethics Committee (2017/1197).

We moved the Fig. 4 (HeLa cell experiment part) to Supplementary Section V as Supplementary Fig. S7. As the application of NIRES, new Fig. 4 & 5 with the elucidative paragraphs are added in main text as below:

We further examine the penetration depth and resolution of NIRES imaging through deep tissue (**Fig. 4**). In this experiment, 4% Tm³⁺ 40% Yb³⁺ co-doped UCNPs are attached behind a 93 μm thick slice of mouse liver tissue, which allows UCNPs diffuse into the tissue slice for super-resolution imaging of single UCNPs from different depth (**Fig. 4a**). Due to the aforementioned strong attenuation for visible emissions (shown in **Fig. 4b** and **Supplementary Fig. S8**), through a 93 μm liver tissue slice, there is only 11.3% of 455 nm emission left in confocal imaging (**Fig. 4c**), and in contrast there are 38.7% of strong signal at 800 nm detectable in both confocal imaging (in red) and NIRES (in purple) super resolution imaging. More encouragingly, a fairly consistent resolution of sub-50 nm has been maintained without any aberration correction through a tissue as deep as 93 μm (**Fig. 4d**). This is because that the wavelength of 980 nm excitation and 800 nm emission minimize the aberration by tissue.

Figure 4. The penetration depth of different emission bands and optical resolution of different imaging modalities at different depth of a liver tissue slice. (a) An illustration of a mouse liver tissue slice with 93 μm thickness. (b) Single particle imaging at different depth in liver tissue. Left, confocal images from 455 nm emission; middle, confocal images from 800 nm emission; right, the corresponding NIREs images. (c) The normalized emission attenuation at different depth through the liver tissue. (d) The corresponding FWHM in (b); The resolutions of NIREs in (d) are 49.6 ± 11.1 nm (0 μm), 42.4 ± 6.2 nm (15 μm), 42.4 ± 7.2 nm (55 μm), 48 ± 7.3 nm (75 μm), 38.18 ± 14.3 nm (93 μm). Detailed data in (d) is shown in **Supplementary Table S2**. Pixel dwell time for confocal and NIREs is 3 ms. The pixel size for confocal and NIREs is 10 nm. The scale bar is 500 nm.

Fig. 5 further examines the resolution power of NIREs in resolving single UCNPs from nanoparticle clusters in deep tissue. **Fig. 5a** shows a bright field image of an 88 μm liver tissue slice, merged with the fluorescence image of four clusters of UCNPs (in red). At this depth, 800 nm confocal microscopy (**Fig. 5b**) does not provide sufficient resolution to image single UCNPs within a diffraction limit area (**Fig. 5c & d**). In contrast, NIREs nanoscopy can clearly resolve single UCNPs by either negative (**Fig. 5e & f**) or positive images (**Fig. 5g & h**). The two typical areas of confocal and NIREs/NIREs+ images, marked by orange square (**Fig. 5c, e & g**) and green square (**Fig. 5d, f & h**) respectively, clearly illustrate the sub-50 nm resolution achieved by NIREs nanoscopy, in resolving single UCNPs separated with 161 nm and 275 nm (**Fig. 5i and j**). It is notable that the resolvable distance of 161 nm is not the limit. For instance, UCNPs with a distance 72 nm can be resolved in **Supplementary Fig. S11**. **Supplementary Fig. S12** provides additional data showing that the best resolutions of 54 nm and 62 nm have been achieved through an 85 μm mouse kidney tissue slice and a 92 μm brain tissue slice, respectively.

Figure 5. NIREs nanoscopy for super resolution imaging of single UCNP through deep mouse liver tissue. (a) Bright field and fluorescence wide field image of UCNP at 88 μm depth inside a mouse liver tissue slice. (b) Confocal images of a $6\ \mu\text{m} \times 6\ \mu\text{m}$ area containing clusters of UCNP. (c) and (d) are the zoom-in confocal images of two areas of interest from (b). (e) and (f) are the raw data of NIREs images of (c) and (d). (g) and (h) are the processed data of NIREs+ images of (e) and (f). (i) Line profiles of UCNP from the confocal image (c), raw NIREs image (e), and positive NIREs image (g). (j) Line profiles of UCNP from confocal image (d), raw NIREs image (f), and positive NIREs image (h). Pixel dwell time for confocal and NIREs is 3 ms. The Pixel size for confocal and NIREs is 10 nm. The scale bar is 8 μm in (a) and 1.5 μm in (b) and 500 nm in (c) – (h). Note that the excitation power density to achieve the best resolution (see **Supplementary Fig. S9 & S10**) in the depth of 88 μm is 5.5 MW cm^{-2} , to compensate the scattered excitation power through the thick tissue.

Fig. 1. The summarized minimum energy densities required by a range of optical probes for deep tissue super-resolution. In order to facilitate the comparison, we normalize the excitation/depletion power to certain pulse period (200 fs) through 100 μm tissue. The red and blue curves show the light effective attenuation coefficient in the tissue¹. 1PE, one-photon excitation; 2PE, two-photon excitation; QD, quantum dots; FP, fluorescence protein; SEMI, semiconductor nanowires segments

From the UV-vis absorption spectra (**Supplementary Fig. S8**), we find that the absorption efficiencies of these tissues are similar to the skin and fat instead of blood. Blood model is more suitable for the animal organ imaging, while skin and fat are ideal for the tissue slice for the super resolution imaging. **We changed the effective attenuation confident curves of blood to skin and fat in Fig. 1. Moreover, we recalculated the minimum energy densities of the optical probes in Fig. 1 with the details in in Supplementary Section I. Energy densities for the probes.**

New **Supplementary Fig. S8-S12 and Table S2** with the elucidative paragraphs are added in main text as below:

To demonstrate the advance of the longer wavelength for deep tissue penetration, we use the UV-Vis spectrophotometer (Cary 60 UV-Vis, Agilent Technologies) to measure the light absorption through 50 μm and 100 μm liver, brain, and kidney tissue slice samples, respectively (**Supplementary Fig. S8**). As can be seen, the extinction rates of these tissues decrease in general with increasing the light wavelength in visible and near-IR region.

Supplementary Figure S8. UV-vis absorption spectra of the 50 μm and 100 μm live, brain, and kidney tissue slice samples, respectively.

	0 μm	15 μm	55 μm	75 μm	93 μm
455 nm	321.2 \pm 15.7 nm	329.8 \pm 24.3 nm	341.5 \pm 18.9 nm	359.4 \pm 14.5 nm	400.7 \pm 16.4 nm
800 nm	445 \pm 10.5 nm	423.3 \pm 27.5 nm	481.7 \pm 34.3 nm	475 \pm 33.5 nm	488.3 \pm 17.6 nm
NIRES	49.6 \pm 11.1 nm	42.4 \pm 6.2 nm	42.4 \pm 7.2 nm	48 \pm 7.3 nm	38.18 \pm 14.3 nm

Supplementary Table S2. FWHM of 455 nm, 800 nm emission confocal and NIRES at different depth of a liver tissue slice

Supplementary Figure S9. Super-resolution scaling Δr of UCNPs as a function of the excitation power at 93 μm depth inside liver tissue. Error bars indicates standard deviations from line profiles of several measurements. Pixel dwell time, 3 ms; pixel size, 10 nm.

Supplementary Figure S10. Resolved two particles with distance below the diffraction limit in 65 μm depth inside liver tissue. (a) Confocal scanned image (2 $\mu\text{m} \times 2 \mu\text{m}$) of the UCNPs sample. (c) The same position obtained by NIRES, with distinct UCNPs that could not be separated by confocal. (d) The positive NIRES sub-diffraction image by (b) subtracted from (a). (d) Cross-section line profile of UCNPs in raw NIRES image (b), subtracted image (c) and confocal image (a). Pixel dwell time, 3 ms; pixel size, 10 nm. Scale bar, 500 nm.

Supplementary Figure S11. Resolved two particles with distance below the diffraction limit. (a) Confocal scanned image (6 $\mu\text{m} \times 6 \mu\text{m}$) of the UCNPs sample. (b) Confocal image enlarge the red dotted square in (a). (c) The same position obtained by NIRES, with distinct UCNPs that could not be separated by confocal. (d) The positive NIRES sub-diffraction image by (c) subtracted from (b). (e) Cross-section line profile of UCNPs in raw NIRES image (c), subtracted image (d) and confocal image (b). Pixel dwell time, 3 ms; pixel size, 10 nm. Scale bar, 1.5 μm in (a); 500 nm in (b) – (d).

Supplementary Figure S12. NIRES images in deep mouse brain and kidney tissue. (a) NIRES image at 92 μm depth inside of brain tissue (upper) and 85 μm depth inside kidney tissue (bottom). (b) The corresponding cross section profile lines. Pixel dwell time, 3 ms; pixel size, 10 nm. Scale bar, 500 nm.

Major action 2:

To clarify the definition of NIRES resolution and the simulation method, we **have added a new Supplementary Section II. Resolution of NIRES nanoscopy**. In this section, we claim the resolution value we used in previous draft is the experimental resolution which is from the convolution between effective PSF of NIRES and the profile of nanoparticle. To avoid the confusion, in the revised draft we define the ‘FWHM of dip in experimental PSF’ is the experimental resolution. The resolution/effective resolution of NIRES, which indicates the best resolution can be achieved by using smaller particle, is calculated through deconvolution between experimental PSF and nanoparticle’s profile. **We further update the Fig. 2c & d** use simulated excitation PSF. The previous Fig. 2c was calculated using experimental measured excitation PSF. **We also update the resolution data in Fig. 3 to the effective resolution.**

The details of Supplementary Section II Resolution of NIRES nanoscopy is shown below:

Section II. Resolution of NIRES nanoscopy

1. Simulation Method

The approximate function of optical resolution in a STED or GSD microscope has been derived, releasing the famous square root law². The full-width at half maximum (FWHM) of STED point spread function (PSF) with fluorophore that contain two energy level (**Supplementary Fig. S1a**) can be represented as:

$$\Delta x = \frac{h_0}{\sqrt{\zeta}} \quad (\text{S3})$$

Here h_0 denotes the FWHM of confocal PSF. $\zeta = I_{STED}^{Max}/I_S$ denotes the saturation factor. I_S is referred to the saturation intensity where the emission intensity decreases to half of maximum. I_{STED}^{Max} represent the maximum amplitude of STED beam profile.

The validity of this function of FWHM in STED actually can be extended for two-photon STED and two-photon negative GSD where the fluorophore is excited by absorbing two photons with energy below its band gap, through modifying the function of saturation intensity as $I_S = \sqrt{k_{BA}/\sigma_{TPA}}$. Here k_{BA} is the carrier transition rate from excited state B to ground state A, σ_{TPA} denotes the molecular cross section with two-photon absorption.

The NIRES nanoscopy in this paper has similar physical process with two-photon negative GSD but may not be able use the existing function of resolution (**Supplementary Equation S3**). UCNPs has much more complex rate transition system (**Supplementary Fig. S1b**) than the aforementioned two-level system, with rate equation shown as:

$$\frac{dn_1}{dt} = -c_1 n_1 n_{S2} + a_{21} w_2 n_2 + a_{31} w_3 n_3 + a_{41} w_4 n_4 + a_{51} w_5 n_5 - k_{41} n_1 n_4 - k_{31} n_3 n_1 - k_{51} n_5 n_1$$

$$\frac{dn_2}{dt} = c_1 n_1 n_{S2} - c_2 n_2 n_{S2} - a_{21} w_2 n_2 + a_{32} w_3 n_3 + a_{42} w_4 n_4 + a_{52} w_5 n_5 + k_{41} n_1 n_4 + 2k_{31} n_1 n_3$$

$$\frac{dn_3}{dt} = c_2 n_2 n_{S2} - c_3 n_3 n_{S2} - (a_{31} + a_{32}) w_3 n_3 + a_{43} w_4 n_4 + a_{53} w_5 n_5 + 2k_{51} n_5 n_1 + k_{41} n_4 n_1 - k_{31} n_3 n_1$$

$$\frac{dn_4}{dt} = c_3 n_3 n_{S2} - c_4 n_4 n_{S2} - (a_{43} + a_{42} + a_{41}) w_4 n_4 + a_{54} w_5 n_5 - k_{41} n_1 n_4$$

$$\frac{dn_5}{dt} = c_4 n_4 n_{S2} - (a_{54} + a_{53} + a_{52} + a_{51}) w_5 n_5 - k_{51} n_1 n_5$$

$$\frac{dn_{S2}}{dt} = P_{980} n_{S1} - w_{S2} n_{S2} - (c_1 n_1 + c_2 n_2 + c_3 n_3 + c_4 n_4) n_{S2}$$

Here we simplify the energy level which involves two energy levels associated with the sensitizer Yb^{3+} ions and five associated levels with the activator Tm^{3+} ions; n_{S1} , n_{S2} , n_1 , n_2 , n_3 , n_4 and n_5 are the populations of ions on energy levels of ${}^2\text{F}_{7/2}$, ${}^2\text{F}_{5/2}$, ${}^3\text{H}_6$, ${}^3\text{H}_5/{}^3\text{F}_4$ and ${}^3\text{F}_{2,3}/{}^3\text{H}_4$ respectively; c_i is the energy transfer ratio between Yb^{3+} on the excited level and Tm^{3+} both on the ground and the intermediate levels; K_{ij} is the cross-relaxation coefficients between the state i and j ; a_{ij} is the branching ratio from energy level i to j ; W_i is the intrinsic decay rate of Tm^{3+} on level i ; P is the absorption rate of Yb^{3+} ; n_3 (${}^3\text{H}_4$) is the excited state used in this paper. Therefore, the resultant carrier number (emission intensity) function of excitation power is significant different with that for two-photon excited two-level system as shown in **Supplementary Fig. S1c**, which further results in a different function of resolution for NIRES nanoscopy.

According to the description in maintext and shown in **Supplementary Fig. S1c** and **Fig. S6c**, 4% Tm^{3+} doped UCNPs have larger onset value than 2% and 3% Tm^{3+} doped UCNPs, and smaller I_{MAX} value than 6% and 8% Tm^{3+} doped UCNPs. Therefore 4% is the optimized doping concentration for NIRES. Note that 4% Tm^{3+} doped UCNPs have a similar power dependent curve with that for two-photon excited two-level system, which indicates that the best resolution they can achieve is same. It is also notable that even though the best resolution for 4% Tm^{3+} will be similar with that for two-photon excited two-level system, 4% Tm^{3+} still can produce much better resolution with limited excitation power as it has much smaller I_S .

Following a similar derivation as in previous works^{2,3}, we define the effective PSF of the NIRES as:

$$\begin{cases} h_{eff}(x) = h_{em}(x) \times h_c(x) \\ h_{em}(x) = \eta(i) \times h_{exc}(x) \end{cases}$$

Here $h_{em}(x)$ is the PSF of emission; $h_{exc}(x)$ is the PSF of excitation beam (donut beam for NIRES); η is excitation power dependent emission intensity curve; $h_c(x)$ is the PSF of the confocal collection system. The FWHM of the intensity dip in h_{eff} represents the resolution for NIRES nanoscopy.

Supplementary Figure S1. Rate transition system of UCNPs. (a) The energy level diagram of two levels system, with excited state B and ground state A. (b) The energy level diagram of Tm^{3+} and Yb^{3+} doped UCNPs. (c) Simulated excitation power dependent emission intensity for two-photon excited two energy level system (labeled as 2 photon), three-photon excited two energy level system (labeled as 3 photon) and UCNPs with 2%, 4% and 8% Tm^{3+} doping. It is noted that the excitation power is normalized to saturation intensity (I_S) where the emission intensity is dropped by half.

2. Experimental Result and influence of particle size

The experimentally measured PSF (h_{exp}) of NIRES is the convolution between the h_{eff} and the spatial profile (h_{UCNP}) of nanoparticle as below:

$$h_{exp} = h_{eff} * h_{UCNP}$$

The deconvolution process on h_{exp} results in a measured h_{eff} , in which the FWHM of the dip represent the resolution of NIRES. In this paper the resolution is calculated through deconvolution of experimental measured PSF.

The theoretical simulation of h_{exp} for different particle size is shown in **Supplementary Fig. S2a**, where the PSF of UCNPs with size 0nm indicates the h_{eff} . The result indicates that larger particle size leads to larger value of the FWHM of h_{exp} (**Supplementary Fig. S2b**) and the dip height of h_{exp} (**Supplementary Fig. S2c**), which in turn offers lower resolution. It is notable that, the FWHM of the dip of h_{exp} can be smaller than the particle's size (**Supplementary Fig. S2b**) when the FWHM of h_{eff} is smaller than the particle's size, which

stems from the donut shape PSF of h_{eff} . If the shape of h_{eff} is a Gaussian function, the FWHM of the PSF after convolution is always larger than the size of particle.

Supplementary Figure S2. Theoretical simulation of image of single UCNP by NIREs. (a) The PSF of UCNP with size varying from 0nm to 50nm. (b) The FWHM of the dip in h_{exp} for UCNPs with different size. (c) The ratio of the height value of the dip with the peak value in h_{exp} for UCNPs with different size. The UCNPs with 4% Tm^{3+} , 20% Yb^{3+} are used in this simulation. The excitation peak intensity is 100 times larger than the saturation intensity of UCNP.

Additional minor changes have been taken on author list, abstract, introduction and conclusion.

Reviewers' Comments:

Reviewer #1 (Remarks to the Author):

This is a new report for nanoscopy by near-infrared emission saturation (NIREs) from the same research group where STED nanoscopy based on UCNPs was reported. I would say this new NIREs is a better version of nanoscopy than the previous one based in STED because it needs only one excitation laser at 980 nm: simpler way to do nanoscopy. Manuscript is well written and data looks convincing, so I would recommend this manuscript can be published in its current form.

We highly appreciate the Reviewer's supportive remarks and comments.

Reviewer #2 (Remarks to the Author):

This paper entitled "Multi-photon near-infrared emission saturation nanoscopy using upconversion nanoparticles" reports a near-infrared emission saturation (NIREs) nanoscopy for super-resolution using lanthanide-doped upconversion nanoparticles. Setting a 980 nm excitation laser to create a doughnut beam and detecting at 800 nm, the authors achieved a spatial resolution of 34.4 nm, benefiting from the easily saturable upconversion luminescence. Intercellular imaging to track multiple nanoparticles is also demonstrated. Although the results are interesting, the novelty of this work is too weak to fulfill the publication requirements of Nature Communications, referring to my comments below. Thus, I do not recommend it for publication. I suggest the authors submit it to a specific journal.

1. The principle of this work is essentially the same with a few works, such as ESSat microscopy (Yang et al. Nature Photonics 2015, 9, 658), "negative" GSD (Rittweger et al. EPL 2009, 86, 14001) and fluorescence emission difference microscopy (FED) (Kuang et al. Scientific Reports 2013, 3:1441). In these techniques, a doughnut-shaped beam is used for excitation and a full-width at half-maximum of the dip at the central of the pattern determines the resolution of microscopy. Emission saturation of fluorophores is the key feature which is used to increase the imaging resolution (Yang et al. Nature Photonics 2015, 9, 658; Zhao et al. Optics Express 2016, 23596), which is also used in this work. In addition, the reviewer notices that use of upconversion nanoparticles for ESSat or FED microscopy, utilizing their emission saturation property, has been also reported by another group recently (Wu et al. Optics Express 2017, 25, 30885). I do not see any conceptual innovation from the current work to fulfill the publication requirement of Nature Communications. I also notice that in the ref. Wu et al. Optics Express 2017, 25, 30885 the authors used 800 nm excitation and detecting at 650 nm. The working wavelengths (980 nm excitation, 800 nm emission) used in the current work are not advantageous.

Though the fundamental principles of the super resolution techniques employing UCNPs is not novel, our revised manuscript offers for the first time a super resolution imaging of a deep tissue with resolution below 50nm, using CW laser. This only became possible due to a careful selection of emission wavelength and doping concentration for UCNPs. This is the first evidence for a super resolution nanoscopy with both excitation and emission in near-infrared (IR) wavelength applied directly for tissue imaging – highly advantageous for damage free biological imaging. Furthermore, the attenuation coefficient for 650 nm is higher than that for 800nm for slice live, brain, and kidney tissue sample as shown in Supplementary Section VI. All mentioned papers have been cited in our original submission and discussed in this revised version.

We have added this to the introduction:

Most recently Wu et al. demonstrated UCNPs can achieve fluorescence emission difference (FED) microscopy⁴, with 172 nm resolution by employing 800 nm excitation under 10 MW cm⁻². However, the lower resolution, the higher excitation power and the 660 nm emission wavelength limit its application in deep tissue super-resolution imaging.

2. The authors compare a series of nanoparticles with different Tm concentrations. I am not a material expert. But to me it is just screening the nanoparticles. It is lack of an in-depth study to construct a connection between nanoparticle engineering and emission saturation properties, thus seeming lack of guiding significance.

We have amended the discussion on the preparation of the nanoparticles, as detailed below.

Similar with GSD or ESSat sub-diffraction imaging approaches,^{5,6} the resolution of NIRES at a certain excitation power density is essentially determined by the emission saturation curve (**Fig. 2b**). In our case, there are three features from the curve affecting the resolution. The first feature is the power point (I_S) to achieve the half value of maximum emission intensity. The second feature is the power point (I_{MAX}) to achieve the maximum emission intensity with fixed I_S , in other words smaller I_{MAX} indicates more superlinear shape of the curvature between I_S and I_{MAX} . The third feature is the onset value of the curve with fixed I_S , with larger onset indicates more underlinear shape of the curvature between 0 and I_S . Lower values of I_S and/or I_{MAX} decrease the size of dark spot in the emission donut pattern, measured by the FWHM value of the dip, thereby enhancing the resolution. Larger onset of the curve offers lower depth of the donut emission PSF to yield better resolution.

The saturation curve of UCNP can be optimized by tuning the doping concentration of emitters (**Supplementary Fig. S6**). UCNPs with lower Tm³⁺ doping concentration can be easily saturated with lower values of I_S and I_{MAX} (**Supplementary Fig. S6**) due to their smaller energy transfer ratio⁷ and resultant lower saturated carrier flow rate, which is proportional to laser induced carrier generation rate. The lower values of I_S and I_{MAX} are favorable towards achieving higher resolution. However, lower Tm³⁺ doped UCNPs have smaller onset of their saturation curve (2% Tm³⁺ in **Supplementary Fig. S6c**), which substantially affects the resolution. Therefore, it is hard for UCNPs with 2% and 3% Tm³⁺ to achieve higher resolution even by increasing excitation power (**Fig. 3**). UCNPs with high Tm³⁺ doping concentration have shaper curvature of their onset of the saturation curves, but the values of I_S and I_{MAX} (**Supplementary Fig. S6b**) are too high because of larger energy transfer ratio⁷ and higher cross-relaxation rate⁸ for high Tm³⁺ doped UCNP, therefore do not benefit to achieve higher resolution. As shown in **Fig. 3**, UCNPs with 6% and 8% Tm³⁺ require extremely high power to achieve the same resolution as that for 4% Tm³⁺ UCNPs, optimized for NIRES (**Supplementary Fig. S1c**). With that, the highest resolution for single nanoparticle imaging is 33.9 ± 12.3 nm (**Fig. 3** insert) at an excitation power density of 4 MW cm⁻². This resolution can be further improved by optimizing the sensitizer concentration or designing a core-shell structure, suggesting a large scope for materials science community to improve the resolution of NIRES nanoscopy. Note that the conventional method of square root law cannot be used to fit the resolution of NIRES due to the unique saturation curve of UCNPs. **Supplementary Section I** provides more details on the fitting method to calculate the theoretical resolution in imaging single UCNP by NIRES (**Supplementary Fig. S2**). **Supplementary Section V Fig. S7** provides more data using NIRES to resolve single UCNPs inside HeLa cell with a resolution of 65 nm in biological environment.

3. The size of nanoparticles used in this study is around 40 nm in diameter. Then the argued resolution of 34.4 nm does not make sense.

This is an important point. Due to the unique shape of donut emission PSF (h_{eff} in in Supplementary Section II), the FWHM of the dip in PSF of experimental NIRES imaging (h_{exp} in Supplementary Section II) can be smaller than the size of nanoparticles for NIRES, when FWHM of the dip in h_{eff} is smaller than the size of nanoparticle (see theoretical simulation in in Supplementary Fig. S2). For that case we can resolved two particles with space smaller than the size of nanoparticle, even though it cannot be achieved in reality. In fact, larger size of particles will result in lower experimental resolution (FWHM of h_{exp}) and higher value of center dip, as h_{exp} is the convolution of h_{eff} with particle's profile. **We added this explanation to the Supplementary Section II.**

4. In addition, these nanoparticles as others like QDs are difficult to modify and functionalize, so they are not advantageous compared to organic biomarkers.

We respectfully disagree here.

Compared to organic dyes, luminescent nanoparticles offer high brightness and photo stability for long-term tracking of single molecules and real-time super-resolution imaging of sub-cellular structures, as the current molecular dyes are often too dim, and suffer from rapid photobleaching. The excitation wavelength and emission wavelength through NIR biological window, demonstrated in this work, show another advance using nanoparticles.

In fact, though compared to dye, significant challenges lie ahead in mastering their complicated surface biochemistry, many recent developments have resulted in a large collection of nanoscopically sized probes with well-defined optical properties for tracking single molecules and super-resolution imaging of sub-cellular structures.

Encouragingly, many recent studies on functionalization and bio-conjugation of UCNPs have shown more promise for specific cell imaging^{9,10,11}, detection of low-abundant biomolecules^{12,13}, cell optogenetics^{14,15,16} and long-term tracking in live cells^{17,18,19,20}. **We have added this to the discussion** as shown below:

The recent progress on functionalization of UCNP, enabling bio-conjugation^{21,10,22,23}, cell optogenetics^{14,15,16} and long-term tracking in cell^{17,18,19,20}, will empower NIRES to track more biological events.

5. The dwell time used is 3 ms, which is much longer than those used for organic biomarkers, typically tens of us. Such long dwell time would limit the imaging speed.

We respectfully disagree here.

To imaging single nanoparticle, we used relatively longer dwelling time in this work. It also shows the advantage of UCNPs being extremely photostable and bright. To our best knowledge, it is impossible to image single “organic biomarkers” (organic dye) by STED/GSD super resolution nanoscopy, because of weak signal and photo-bleaching issue.

Scanning at high speed to detect UCNPs is feasible, as recently demonstrated by Zhan et al.⁷, in which work 100 μ s dwelling time used to realize 80 nm resolution for imaging subcellular structures. High speed imaging can also be realized by using an array of multiple donut beams²⁴.

Reviewer #3 (Remarks to the Author):

The authors present results on multi-photon near-infrared emission saturation nanoscopy using upconversion nanoparticles. The upconversion nanoparticles used in their paper allow multi-photon excitation with low laser intensity. Moreover, both the excitation and emission light are in the near-infrared wavelength range, potentially facilitating the imaging of thick samples. On top of the multiphoton excitation, a technique similar to ESSat or GSD is applied to break the diffraction limit and to achieve super-resolved imaging. The approach and the results presented here are very interesting and promise a simple and powerful way of performing super-resolution imaging in deep tissue. Considering the potential impact of this work, I recommend publication. However, there are a few issues that the authors should take into account:

1. It is not clear how the light intensities in Figure 1 are calculated for various imaging modalities. Please add more information.

We appreciate the review's comment. The initial setting conditions are important for normalized calculation, and we apologized for lack of details. **We have added Supplementary Section I Energy densities for the probes**, summarised the key parameters of these probes for deep tissue imaging to clarify the calculation of energy intensity, as shown below:

Section I. Energy densities for the probes

The most common probes to achieve STED^{25,26,8} and GSD²⁷ (**Supplementary Table S1**) includes fluorescent proteins, quantum dots, semiconductor nanowires and UCNPs. **Supplementary Table S1** summaries the key parameters for these probes. To compare the maximum laser induced energy dosage, required by different probes to achieve nanoscopy through deep tissue, we calculated the energy density (I_Q) of both excitation and depletion laser during excitation time of 200 fs through 100 μm skin tissue.

We can calculate the required I_Q according to:

$$R_{tr} = e^{-\alpha_\lambda l} \quad (\text{S1})$$

$$I_Q = \frac{Pr_o t_\tau}{Aft_p R_{tr}} \quad (\text{S2})$$

Where R_{tr} is the transmittion ratio of an electromagnetic wave penetrating a material (Beer's law); P is the beam power; r_o is the loss rate of the objective lens; t_τ is the exposure time; A is the area of the focused laser spot; f is the pulse frequency; t_p is the pulse duration; α_λ is the attenuation coefficient; λ is the wavelength; l is the path length. Note that the value of f and t_p is 1 for CW laser. The loss rate of the laser through the objective lens is based on our current system ($r_o = 0.43$), and the area of the focused laser spots are $A_{\text{gau}} = 3.76 \times 10^{-9} \text{ cm}^2$ for Gaussian beam and $A_{\text{dou}} = 7.81 \times 10^{-9} \text{ cm}^2$ for Doughnut beam. The exposure time is 200 fs, and the path length is 100 μm .

Supplementary Table S1 in the revised manuscript). Key parameters of various imaging modalities for deep tissue

Nanoscope	Probe	λ_{ex} (nm)	λ_{dep} (nm)	f (MHz)	Pulse duration t_p (ps)	Power intensity I (mW)	Energy intensity I_Q (J cm ⁻²)
STED ²⁵	1PE-QD	628	-	38	1.2	0.05	7.9×10^{-5}
		-	775	38	1200	150	
STED ²⁶	2PE-FP	850	-	76	0.13	2.7	9.4×10^{-3}
		-	580	76	200	4.4	
STED ⁸	MP-UCNP	980	-	CW	-	1	7.6×10^{-7}
		-	808	CW	-	40	
GSD ²⁷	1PE-SEMI	700	Non	80	5	5	2.5×10^{-4}
NIRES	MP-UCNP	980	Non	CW	-	75	1.2×10^{-6}

Where 1PE, one-photon excitation; 2PE, two-photon excitation; MP, multi-photon excitation; QD, quantum dots; FP, fluorescence protein; SEMI, semiconductor nanowires segments

We have rewritten the paragraph of “To meet the requirement of super-resolution imaging in deep tissue, UCNP requires the smallest laser power for deep tissue super-resolution imaging” to a paragraph as shown below:

To meet the requirement of super-resolution imaging through deep tissue, we first calculate and examine the minimum excitation/depletion energy intensity (J cm^{-2}) for fluorescent proteins²⁶, quantum dots²⁵, semiconductor nanowires segments²⁷ and UCNPs⁸ used in STED and GSD approaches (Fig. 1). The normalized calculation of energy densities for deep tissue imaging is shown in Supplementary Section I and key parameters of these probes are summarized in Supplementary Table S1. The tissue attenuates more power for shorter wavelength, which requires large power in visible range to achieve high resolution, if using one-photon excitation (1PE) scheme. That is why two-photon excitation (2PE) in NIR range is the commonly used method. But multi-photon probes have extremely small absorption cross-sections, thereby requiring even higher excitation power. In contrast, taking both advantages of NIR excitation wavelength and large absorption cross-section, UCNPs provide a potential solution by only requiring small laser power through the deep tissue.

2. Please describe briefly the diagram in Figure 2a.

To explain the photon upconverting process shown in Fig 2a, we updated Fig.2 and rewrote the caption with more details for the diagram as below:

Figure 2. The principle of NIREs nanoscopy using UCNP as multi-photon probe for deep tissue imaging. (a) The simplified energy levels and upconversion process of Yb³⁺ and Tm³⁺ co-doped UCNPs. The sensitizer Yb³⁺ ions the photon upconversion process comprises a linearly and sequential absorption of 980 nm excitation. Due to the multiple long-lived real intermediate states, the energy is stepwise transferred onto the scaffold energy levels of emitters Tm³⁺, eventually facilitate multiphoton upconversion emission, including emissions from the four photon upconversion excited state ¹D₂ (455 nm), three photon state ¹G₄ (470 nm), and two photon excited state ³H₄ (800 nm). (b) The saturation intensity curve of the 800 nm

emissions from UCNPs (40 nm NaYF₄: 20% Yb³⁺, 4% Tm³⁺) under 980 nm excitation. (c) Cross-section profiles of the saturated upconversion emission of UCNPs at four different excitation powers of 0.1 MW cm⁻², 0.4 MW cm⁻², 1 MW cm⁻² and 3 MW cm⁻². (d) The simulated ‘negative’ contrast images of the cross-section profiles of a single UCNP. Pixel size, 10 nm. Scale bar is 500 nm.

3. Line 117: “the dots of experimental data are fitted well to the simulation results”. It is not clear what type of simulation was performed and how the data were fitted. Please provide more details.

The simulation method is important, and we apologized for lack of details in our original submission. **We have added a Supplementary Section II Resolution of NIRES nanoscopy**, to clarify the simulation method.

4. In ESSat or GSD, the resolution is proportionally to the inverse of the squared root of the depletion intensity; what is the quantitative relationship between the resolution and the laser intensity in NIRES?

In the new Supplementary Section IV, we have discussed that the excitation power dependent resolution in NIRES. To summarize, the UCNP have much complex energy level system compared with traditional two-level system (Supplementary Fig. S1b). The available carrier number is not in the format of $N_A = k_{BA} / (k_{BA} + \sigma I^N)$ any more, here k_{BA} is the carrier transition rate from excited state B to ground state A, σ denotes the molecular cross section with N-photon absorption. Therefore, the power dependent emission curve for two-photon level (emitting 800nm) in UCNP is much different from that for two-level two photon system (Supplementary Fig. S1c), which results in slightly different trend of change of resolution with respect to different power. The relationship cannot be simplified as the inverse of the squared root law. **The simulated relationship has been shown in revised Fig. 3**, but it is hard to be fitted into a meaningful formula.

Figure 4. Super-resolution scaling Δr of UCNP (NaYF₄:20% Yb³⁺, x% Tm³⁺, ~40 nm in diameter; x = 2, 3, 4, 6 and 8) as a function of the excitation power (intensity). The dots of experimental data are fitted well to the simulation results (solid lines; Supplementary Fig. S5). Error bars indicate standard deviations from line profiles of several measurements. Insets, NIRE images of 8% Tm³⁺ doped UCNP at 5.5 MW cm⁻² (left), 4% Tm³⁺ one at 4 MW cm⁻² (middle), and the corresponding cross section profile lines (right). Pixel dwell time, 3 ms; pixel size, 10 nm. Scale bar is 500 nm. The series of NaYF₄: 20% Yb³⁺, x% Tm³⁺ UCNP (x=2, 4, 6, and 8) are controlled synthesized and characterized by following previously reported methods^{8,28}, and shown in Supplementary Section III Fig. S4.

- Line 135: “further increase of excitation power results in a high background of dip The value of center dip of the doughnut distribution is often above zero”. Please give some quantitative values. What is the ratio of the intensity at the doughnut center to the max intensity of the beam? It is important to evaluate this value carefully, especially considering this ratio could be further compromised when imaging through thick samples. Is this ratio constant when increasing the laser power? In Figure S5, it seems that the intensity at the center dip is more pronounced for UCNP with lower Tm³⁺ concentration. If it is true, please explain it.

The ratio value of the intensity at the doughnut center to the max intensity of the beam (I_c) is 1.39%, which is measured by detecting the scattering intensity of a 100 nm gold bead during a confocal scanning process. **We have added the PSF of the excitation beam of NIRE in XY and XZ plan, and the center cross section profile with $I_c=1.39\%$ to Supplementary Fig. S3.** Because the pattern of the PSF is obtained by the linear reflection of the gold particle, I_c is a constant when changing the laser power. It is true that the center dip is more pronounced for UCNP with lower Tm³⁺ concentration. As mentioned in the revised maintext, UCNP with lower Tm³⁺ doping concentration has slower slope of the curvature between excitation power of 0 and I_s (Supplementary S6c), which yield higher depth of the dip in donut emission PSF and lower resolution.

The revised Supplementary Fig. S3 and its caption is shown below:

Supplementary Figure S3. Experimental setup for NIREs nanoscopy (SMF, single-mode fiber; MMF, multi-mode fiber; L1, collimation lens; L2, collection lens; HWP, half-wave plate; QWP, quarter-wave plate; PBS, polarized beam splitter; VPP, vortex phase plate; M, mirror; FM, flexible mirror; DM, dichroic mirror; OL, objective lens; BPF, band pass filter; SPAD, single-photon avalanche diode; CCD, charge coupled device). The dotted portion is designed for auxiliary confocal with two flexible mirrors to bypass the VPP in the main optical path. Inset shows point spread function (PSF) of the NIREs is measured by scattering of a 100 nm gold bead in reflection (path not shown). The I_c (ratio value of the intensity at the doughnut center to the max intensity of the beam) is measured as 1.39%. Scale bars: 500nm.

6. Line 157: “The low saturation intensity at intermediate excited states, as upconversion emissions at the peripheral areas away from the doughnut center are ready saturated”. It is not clear what message this sentence is trying to deliver. Please rephrase the sentence.

We accept the review’s claim and **rewrite the paragraph** of “The low saturation intensity at NIREs from the “confocal image”.” as below:

Same as SAC method²⁹, higher excitation power will raise up the dip in the PSF of emission for NIREs to the maximum point of PSF according to the saturation curve (**Fig. 2b**), thereby switching NIREs into a confocal microscopy obtaining “confocal image”. The subtraction of the “confocal image” with respective to negative NIREs image will further provide a positive NIREs image.

Reference

1. Smith, A. M., Mancini, M. C. & Nie, S. Bioimaging: Second window for in vivo imaging. *Nat. Nanotechnol.* **4**, 710–711 (2009).
2. Hell, S. W. Toward fluorescence nanoscopy. *Nat. Biotechnol.* **21**, 1347–1355 (2003).
3. Zhao, G., Kuang, C., Ding, Z. & Liu, X. Resolution enhancement of saturated fluorescence emission difference microscopy. *Opt. Express* **24**, 23596–23609 (2016).
4. Wu, Q., Huang, B., Peng, X., He, S. & Zhan, Q. Non-bleaching fluorescence emission difference microscopy using single 808-nm laser excited red upconversion emission. *Opt. Express* **25**, 30885 (2017).
5. Rittweger, E., Wildanger, D. & Hell, S. W. Far-field fluorescence nanoscopy of diamond color centers by ground state depletion. *EPL* **86**, 14001 (2009).
6. Han, K. Y., Kim, S. K., Eggeling, C. & Hell, S. W. Metastable dark states enable ground state depletion microscopy of nitrogen vacancy centers in diamond with diffraction-unlimited resolution. *Nano Lett.* **10**, 3199–3203 (2010).
7. Zhan, Q. *et al.* Achieving high-efficiency emission depletion nanoscopy by employing cross relaxation in upconversion nanoparticles. *Nat. Commun.* **8**, 1058 (2017).
8. Liu, Y. *et al.* Amplified stimulated emission in upconversion nanoparticles for super-resolution nanoscopy. *Nature* **543**, 229–233 (2017).
9. Yb, U. N. *et al.* Immunolabeling and NIR-Excited Fluorescent Imaging of HeLa Cells by Using NaYF₄:Yb,Er Upconversion Nanoparticles. *ACS Nano* **3**, 1580–1586 (2009).
10. He, H. *et al.* Bispecific Antibody Functionalized Upconversion Nanoprobe. *Anal. Chem.* **90**, 3024–3029 (2018).
11. Shi, Y. *et al.* Stable Upconversion Nanohybrid Particles for Specific Prostate Cancer Cell Immunodetection. *Sci. Rep.* **6**, 1–12 (2016).
12. Chen, Y. *et al.* Exonuclease III-Assisted Upconversion Resonance Energy Transfer in a Wash-Free Suspension DNA Assay. *Anal. Chem.* **90**, 663–668 (2018).
13. Lu, J., Paulsen, I. T. & Jin, D. Application of exonuclease III-aided target recycling in flow cytometry: DNA detection sensitivity enhanced by orders of magnitude. *Anal. Chem.* **85**, 8240–8245 (2013).
14. Chen, S. *et al.* Near-infrared deep brain stimulation via upconversion nanoparticle – mediated optogenetics. *Science*. **359**, 679–684 (2018).
15. Yadav, K. *et al.* Targeted and efficient activation of channelrhodopsins expressed in living cells via specifically-bound upconversion nanoparticles. *Nanoscale* **9**, 9457–9466 (2017).
16. He, L. *et al.* Near-infrared photoactivatable control of Ca²⁺-signaling and optogenetic immunomodulation. *Elife* **4**, 1–25 (2015).
17. Wang, F. *et al.* Microscopic inspection and tracking of single upconversion nanoparticles in living cells. *Light Sci. Appl.* **7**, 18006–18007 (2018).
18. Nam, S. H. *et al.* Long-term real-time tracking of lanthanide ion doped upconverting nanoparticles in living cells. *Angew. Chemie* **50**, 6093–6097 (2011).
19. Jo, H. L. *et al.* Fast and background-free three-dimensional (3D) live-cell imaging with lanthanide-doped upconverting nanoparticles. *Nanoscale* **7**, 19397–19402 (2015).
20. Bae, Y. M. *et al.* Endocytosis, intracellular transport, and exocytosis of lanthanide-doped upconverting nanoparticles in single living cells. *Biomaterials* **33**, 9080–9086 (2012).
21. Duong, H. T. T. *et al.* Systematic investigation of functional ligands for colloidal stable upconversion nanoparticles. *RSC Adv.* **8**, 4842–4849 (2018).
22. Sun, Y. *et al.* A supramolecular self-assembly strategy for upconversion nanoparticle bioconjugation. *Chem. Commun.* **54**, 3851–3854 (2018).
23. Ren, W. *et al.* Anisotropic functionalization of upconversion nanoparticles. *Chem. Sci.* **9**, 4352–4358 (2018).
24. Chmyrov, A. *et al.* Nanoscopy with more than 100,000 ‘doughnuts’. *Nat. Methods* **10**, 737–740 (2013).
25. Hanne, J. *et al.* STED nanoscopy with fluorescent quantum dots. *Nat. Commun.* **6**, 7127 (2015).
26. Li, Q., Wu, S. S. H. & Chou, K. C. Subdiffraction-limit two-photon fluorescence microscopy for GFP-tagged cell imaging. *Biophys. J.* **97**, 3224–3228 (2009).
27. Oracz, J. *et al.* Ground State Depletion Nanoscopy Resolves Semiconductor Nanowire Barcode Segments at Room

Temperature SI. *Nano Lett.* **17**, 2652–2659 (2017).

28. Ma, C. *et al.* Optimal Sensitizer Concentration in Single Upconversion Nanocrystals. *Nano Lett.* **17**, 2858–2864 (2017).

29. Zhao, G. *et al.* Saturated absorption competition microscopy. *Optica* **4**, 633 (2017).

Reviewers' comments:

Reviewer #2 (Remarks to the Author):

There is no conceptual innovation and optical technological improvements in the present work, compared to previous publications as the reviewer previously commented (Rittweger et al. EPL 2009, 86, 14001; Kuang et al. Scientific Reports 2013, 3:1441; Yang et al. Nature Photonics 2015, 9, 658; Zhao et al. Optics Express 2016, 23596; Wu et al. Optics Express 2017, 25, 30885). The authors achieved better resolution by employing nanoparticles which have better emission saturation properties. In the revised version, the authors included deep tissue super-resolution imaging experiments for trying to prove the power of their technique in deep tissue imaging (a ~ 50 nm resolution at a depth of 93 μm was alleged), which adds value to this field. However, the MS cannot justify its publication in the present form in the esteemed Journal Nature Communications, which has high requirement on novelty and importance. I hope the authors could address my comments below to improve their MS.

1. In the revised version, when trying to highlight the importance of their work, the authors claim in the rebuttal letter that the realization of deep tissue imaging only became possible due to a careful selection of emission wavelength and doping concentration for the UCNPs. However, the level of 100 μm depth is relatively shallow and comparable resolution has been demonstrated with visible/short wavelengths even in living brain slices in some reports (e.g., Urban et al., Biophys. J. 101, 1277-1284, 2011). The 93- μm tissue imaging did not demonstrate the advantage of NIR wavelengths that the author claimed. Much larger depth microscopic imaging would be preferable.
2. In the deep tissue imaging, the tissue samples were treated to remove blood content which would highly affect the imaging quality even for NIR light. Actually, blood is the major origin of contrast for some deep tissue imaging technique, e.g., NIR light enabled optoacoustic imaging. On one hand, the authors emphasize the advantages of NIR wavelength with less absorption for deep-tissue imaging, but on the other hand, the authors used blood (strong absorber)-removed tissues to do experiments, which seems not logical. These treatments could not prove the power of their technique in deep living specimens imaging. It would be more meaningful if the authors carry on experiments directly on fresh slices, e.g., Urban et al., Biophys. J. 101, 1277-1284, 2011, Urban et al., J. Biophotonics, 11, e201700171, 2018. The authors are accordingly suggested to perform UV-vis absorption measurement for fresh slices with blood.
3. The imaging resolution of Fig. 5(a) is poor and should be improved. The authors are suggested to provide photographs for the tissue slices imaged. In addition, the authors should provide super-resolution imaging for continuous patterns rather than several discrete sites to demonstrate their imaging capacity. Such thin tissue slices are typically discontinuous, so some particles are possibly not covered by any tissue.
4. The authors claim that a resolution of 38.2 ± 14.3 nm was achieved through 93 μm thick tissue. The standard deviation is so big that such a claim becomes meaningless. In Figure 4(d), the resolution becomes better at larger depth, which is nonsensical. This implies that their claimed resolution is not reliable. The reviewer suggests the authors to use a more quantitative approach to evaluate the resolution of their imaging.
5. The resolution through 93 μm thick tissue is surprisingly close to the average resolution without tissue (34.4 nm). Because of specimen-induced aberrations, it is a challenge for nanoscopy to maintain consistent lateral resolution in 3D tissue (Gould et al., Opt. Express 20, 20998-21009 (2012)). With advanced optical techniques, e.g., using a hollow Bessel beam (Laser Photonics Rev. 10, 147-152, 2016), relatively consistent lateral resolution can be achieved. Such consistent resolution (w/wo tissue), however, would be very challenging without any advances in the optical setup. Theoretical analysis could verify the experimentally achieved results. The authors are encouraged to provide a theoretical calculation about the imaging resolution change versus depth, considering how the donut-shape beam propagates and how the focusing evolves in the tissue by quantitatively

addressing specimen-induced light (ex/em) absorption and scattering.

6. In supplementary Section II. The authors used a rate equation model to study the relationship between the emission intensity and excitation intensity for nanoparticles with different T_m concentrations. However, they did not provide any parameter values used in the simulations with suitable justifications. Such justification is apparently imperative to make the simulated results meaningful.

7. Fig. 2(b): The authors need to provide a convincing explanation to the emission intensity decrease with increasing excitation intensity after reaching a threshold, correlating to their simulations. Their simulated results do not predict such remarkable intensity decrease (supplementary Fig. S1).

8. Lines 107-123, pages 5-6: The authors used the "onset value" of the emission saturation curve to have discussion, which is too vague. The "onset value" should be quantified quantitatively. The resolution in ESSat, GSD-like, or FED microscopy is fundamentally related with the emission saturation properties of the probes. By modeling the relationship between the excitation intensity and emission intensity, it is very straightforward and thus kind of a routine to quantify the imaging resolution. The authors should perform resolution analysis more quantitatively, as done in previous papers.

9. In Figure 4, the authors claimed that they were doing single-particle imaging. Is there any evidence for that it is single particle?

10. The authors mentioned several times that their method is analogous to, "'negative' Ground State Depletion (GSD-like) mode" (line 59, page 2), "Similar to the principles of ESSat or 'negative' GSD" (line 97-98, page 4), and "Similar with GSD or ESSat sub-diffraction imaging approaches...". The authors are suggested to avoid ambiguity and to describe their method more accurately by clarifying if their technique is GSD, 'negative' GSD or 'negative' GSD-like.

Reviewer #3 (Remarks to the Author):

I believe that the points raised in the previous round of review have been satisfactorily addressed in the revised manuscript. Therefore I recommend publication.

Our detailed responses to the referee #2's comments (in blue), the actions taken (in bold black) and the text change in paper (in red) below:

Reviewers' Comments:

Reviewer #2 (Remarks to the Author):

There is no conceptual innovation and optical technological improvements in the present work, compared to previous publications as the reviewer previously commented (Rittweger et al. EPL 2009, 86, 14001; Kuang et al. Scientific Reports 2013, 3:1441; Yang et al. Nature Photonics 2015, 9, 658; Zhao et al. Optics Express 2016, 23596; Wu et al. Optics Express 2017, 25, 30885). The authors achieved better resolution by employing nanoparticles which have better emission saturation properties. In the revised version, the authors included deep tissue super-resolution imaging experiments for trying to prove the power of their technique in deep tissue imaging (a ~50 nm resolution at a depth of 93 um was alleged), which adds value to this field. However, the MS cannot justify its publication in the present form in the esteemed Journal Nature Communications, which has high requirement on novelty and importance. I hope the authors could address my comments below to improve their MS.

Reading through this reviewer's new comments, we think the reviewer #2 still misses the main points of our work. In our original submission, we report that the saturation effect of UCNPs' intermediate state emissions plays an essential role in achieving a new mode of super resolution nanoscopy, with both excitation and emission in NIR range, as the key for deep tissue super-resolution imaging. This provides a large scope in design and controlled synthesis of UCNPs to improve their optical properties, towards high resolution. Among above references, only *Wu et al's* work is relevant to this work in the field of using UCNPs for emission saturation super resolution nanoscopy, but Wu et al didn't investigate the key points covered by our paper. Plus, Wu's paper was published in Optics Express in late 2017 when our manuscript was submitted to *Nature Communications*, so our submission is independent to Wu's paper.

In our last revision, we add some significant amount of new experiments to demonstrate its power in deep tissue super-resolution imaging of single UCNPs. To our best knowledge, none of other luminescent nanoparticles and single molecule probes has been reported to achieve this level of sensitivity, resolution and penetration depth. Again, this is another important point this reviewer #2 missed.

We would like to acknowledge this reviewer for his/her time spent on reviewing our revision and additional questions to improve our work, therefore we have tried our best by providing some additional data and revising the manuscript for better clarity.

The current version broadly covers areas, includes super resolution imaging of single nanoparticles, simplified optical engineering and setups, controlled synthesis and comprehensive characterizations of nanoparticles, and deep tissue imaging

experiments. After carefully consideration, majority of the recommended references and questions raised by this reviewer are either not so relevant or marginal to the core of our paper, we therefore did not follow some of his/her new suggestions.

Our point-by-point responses are provided below:

1. In the revised version, when trying to highlight the importance of their work, the authors claim in the rebuttal letter that the realization of deep tissue imaging only became possible due to a careful selection of emission wavelength and doping concentration for the UCNPs. However, the level of 100 μm depth is relatively shallow and comparable resolution has been demonstrated with visible/short wavelengths even in living brain slices in some reports (e.g., Urban et al., *Biophys. J.* 101, 1277-1284, 2011). The 93- μm tissue imaging did not demonstrate the advantage of NIR wavelengths that the author claimed. Much larger depth microscopic imaging would be preferable.

We respectfully disagree here.

First, our new Figure 4 clearly shows the advantage of NIR probes in achieving super resolution imaging of a single small nanoparticle, in which regime, the single nanoparticle probe has limited emission intensity and its visible emission decrease seriously to about 10% of its original brightness through a 93 μm thick tissue.

Second, the purpose of Urban et al's work, using visible wavelength STED method, is to image the cell structures, in which regime, the emission was collected from a large number of probes. Therefore, the bright emission can still be detected after serious scattering and absorption from thick tissue.

Moreover, from the level of brightness of a single UCNP emitting at 800 nm, 40% of the original emissions detectable through a 93 μm tissue, we estimated the limit in imaging depth to detect a single nanoparticle using our technique should be around 175 μm . Unfortunately, this experiment is currently limited by the working distance of the objective lens (around 100 μm) used in our current setup.

We added the following discussion paragraph in our revised version at lines 172-176 in page 8: “From the high brightness (20000 photon counts per second, in contrast to the low detection background of 1000 photon counts per second) of a single UCNP emitting at 800 nm through a 93 μm tissue, we estimate the limit in imaging depth to detect a single nanoparticle by NIREs should be around 175 μm . An objective lens with longer working distance (around 100 μm used in our current setup) should be used to confirm this limit in future works.”

2. In the deep tissue imaging, the tissue samples were treated to remove blood content which would highly affect the imaging quality even for NIR light. Actually, blood is the major origin of contrast for some deep tissue imaging technique, e.g., NIR light enabled optoacoustic imaging. On one hand, the authors emphasize the advantages of NIR wavelength with less absorption for deep-tissue imaging, but on the other hand, the authors used blood (strong absorber)-removed tissues to do experiments, which seems not logical. These treatments could not prove the power of their technique in deep living specimens imaging. It would be more meaningful if the authors carry on

experiments directly on fresh slices, e.g., Urban et al., *Biophys. J.* 101, 1277-1284, 2011, Urban et al., *J. Biophotonics*, 11, e201700171, 2018. The authors are accordingly suggested to perform UV-vis absorption measurement for fresh slices with blood.

Thanks for recommending the two references. However, we find the “brain tissue” used in both Urban et al. in 2011 and Urban et al. in 2018 are not “fresh”. It is also an *in vitro* blood-removed tissue section cultured in medium and solution without blood when the slice was imaged. The specific details, from the two reference, are extracted and shown below:

In Urban *et al.*, *Biophys. J.* 101, 1277-1284, 2011, the authors wrote “Organotypic hippocampal slice cultures Hippocampal slices (350 μm thick) were prepared from postnatal day 5–7 wild-type C57BL/6 mice, embedded in a plasma clot on 0.14 mm thick glass coverslips, and incubated in a roller incubator at 35°C”, “The age of the slice cultures used in the experiments ranged from 12 to 42 days in vitro after the preparation.” and “Slices were imaged for a baseline period (typically acquiring two time points) before they were subjected to the modified ACSF solutions. Image stacks were taken typically every 5–10 min. The ACSF that was designed to induce chemical long-term potentiation (LTP) contained (in mM) NaCl 99, KCl 5, MgCl₂ 0.1, CaCl₂ 5, glucose 24, TEA-Cl 25, NaH₂PO₄ 1.25, and NaHCO₃ 26, and was carbogenated to maintain a pH of 7.4. This modified ACSF solution was washed in for a period of 7–9 min before it was washed out by the standard ACSF solution.”

In Urban *et al.*, *J. Biophotonics*, 11, e201700171, 2018, the authors wrote “For cortical and hippocampal imaging, acute brain slices (300 μm -thick) were prepared from postnatal day 23-40 Thy1-eGFP male and female mice. Mice were deeply anesthetized by isoflurane inhalation and sacrificed. Brains were placed into ice-cold, continuously carbogenated (95% O₂/ 5% CO₂), artificial cerebrospinal fluid (ACSF) containing (in mM) 127 NaCl, 2.5 KCl, 25 NaHCO₃, 1.25 NaH₂PO₄, 2.0 CaCl₂, 1.0 MgCl₂, and 25 Glucose (osmolality ~310 mOsm/L).

In this work, we used a standard approach to treat the tissue, and as shown in Figure S8, the blood dry mark can be seen in the tissue. This is because that generally it is hard to remove all blood inside tissue slide. We didn’t observe that these blood mark affected the imaging quality.

We respectfully disagree that it is necessary to carry on fresh sample in this work. Again, the purpose of our work is to demonstrate that NIRES method is powerful due to the excitation and emission wavelength, strong emission and favourable saturation intensity of upconversion nanoparticles. Even though the absorption curve of fresh tissue can be slightly different from the blood removed sample, the transmission of 800 nm and 980 nm is still better than visible wavelength, and upconversion nanoparticles are unique single molecule probes for this purpose.

3. The imaging resolution of Fig. 5(a) is poor and should be improved. The authors are suggested to provide photographs for the tissue slices imaged. In addition, the authors should provide super-resolution imaging for continuous patterns rather than several discrete sites to demonstrate their imaging capacity. Such thin tissue slices are typically discontinuous, so some particles are possibly not covered by any tissue.

First, the bright-field image (Figure 5a) exactly shows that the tissue slice is non-transparent with a lot of scattering under white light illumination.

Second, we appreciate the reviewer for suggesting the photographs of tissue slices. **We added photographs of tissue slices in the Supplementary Information section as Fig. S8, as below:**

Supplementary Figure S8. Photographs of different mouse tissue slices on glass slides. (a) 50 μm , 100 μm , and 150 μm brain tissue slices. (b) 50 μm , 100 μm , and 150 μm kidney tissue slices. (c) 50 μm , 100 μm , and 150 μm liver tissue slices. (d) Kidney and liver tissue with blood slices. (e) Zoom-in image of the blue square region of the tissue slice shown in (c). (f) Zoom-in image of the red square region of the tissue slice shown in (d).

Moreover, we confirmed that the single nanoparticles were all covered by the tissue slice in our experiment, judging from their emission intensities as shown in Figure 4b, c. In Figure 4c, the emission intensity was the value from ten different nanoparticles in different lateral positions. The variation in emission intensities was small, indicating all the nanoparticles covered by a fairly uniform thickness of tissue slice, otherwise the intensity of a single nanoparticle will dramatically increase. This experiment also shows the importance of controlled synthesis of intensity uniform single upconversion nanoparticles.

4. The authors claim that a resolution of 38.2 \pm 14.3 nm was achieved through 93 μm thick tissue. The standard deviation is so big that such a claim becomes meaningless. In Figure 4(d), the resolution becomes better at larger depth, which is nonsensical. This implies that their claimed resolution is not reliable. The reviewer suggests the authors to use a more quantitative approach to evaluate the resolution of their imaging.

We appreciate the review's comment. **We used a conservative number of “sub 50 nm” in abstract.**

Our technology is reliable. The resolution reported here is the highest resolution with optimized excitation power density. The best resolution values in different depth vary because of the difference in tissue aberration. This effect was generally observed in deep tissue imaging. For example, in the reference (Biophys. J. 2011) suggested by this reviewer, Urban et al.¹ showed a sub 75 nm resolution with tissue thickness 0 to 80 μm in figure 2b, where the resolution at 78 μm depth is better than that at 22 μm depth.

We measured the resolution through image deconvolution of the measured point spread function from each single nanoparticle. This approach is common and standard in evaluating the resolution achieved in nanoscopy.

5. The resolution through 93 μm thick tissue is surprisingly close to the average resolution without tissue (34.4 nm). Because of specimen-induced aberrations, it is a challenge for nanoscopy to maintain consistent lateral resolution in 3D tissue (Gould et al., Opt. Express 20, 20998-21009 (2012)). With advanced optical techniques, e.g., using a hollow Bessel beam (Laser Photonics Rev. 10, 147-152, 2016), relatively consistent lateral resolution can be achieved. Such consistent resolution (w/wo tissue), however, would be very challenging without any advances in the optical setup. Theoretical analysis could verify the experimentally achieved results. The authors are encouraged to provide a theoretical calculation about the imaging resolution change versus depth, considering how the donut-shape beam propagates and how the focusing evolves in the tissue by quantitatively addressing specimen-induced light (ex/em) absorption and scattering.

Note that for super-resolution nanoscopy using “donut” beam, such as STED, GSD and NIREs, the aberration induced decrease in lateral resolution can be compensated by extra excitation power. More specifically the PSF of excitation beam will be distorted by deep tissue induced aberration, which result in larger FWHM in emission's PSF (lower resolution). However, with increased excitation power, the FWHM in emission's PSF can be reduced, enabling compensation in resolution. This compensation approach has been commonly used in STED and GSD. For instance, Urban et al.¹, used different excitation power to maintain a resolution of sub 75 nm through brain tissue slide. Here we achieved this compensation in super resolution imaging of single nanoparticles.

To clarify the process, we added “by increasing excitation power to compensate the aberration induced distortion on excitation PSF¹,” to line 147 in page 6, and changed the sentence of “This is because that the wavelength of 980 nm excitation and 800 nm emission minimize the aberration by tissue.” to “The lower refractive index^{2,3} of tissue for NIR light results in less aberration than that for visible beam, which also contributes to the high resolution achieved by NIREs for single nanoparticle imaging.” in main text at line 149 to 150 in page 6.

Other sophisticated methods such as Bessel beam and AO device can be used to compensate the aberration, and will be used in our future work, but here are beyond the scope of this work.

We disagree to add scattering simulation, as the scattering from the tissue is random. There is no point to make some artificial parameter to distort the PSF then compensate it back. The effect that how a thick tissue affects the resolution of NIRES can be directly found by comparing the power dependent resolution inside (Figure S10) and outside deep tissue (Figure 4).

6. In supplementary Section II. The authors used a rate equation model to study the relationship between the emission intensity and excitation intensity for nanoparticles with different Tm concentrations. However, they did not provide any parameter values used in the simulations with suitable justifications. Such justification is apparently imperative to make the simulated results meaningful.

We added the simulation parameters in Supplementary Table S3. This table is shown as below:

Supplementary Table S3. The values of key constants and rate parameters used in the simulations⁴.

$w_2 (s^{-1})$	$w_3 (s^{-1})$	$w_4 (s^{-1})$	$w_5 (s^{-1})$	$w_{s2} (s^{-1})$
63000	20000	15000	33000	8000
a_{51}	a_{52}	a_{53}	a_{54}	
0.24	0.23	0.2	0.33	
a_{41}	a_{42}	a_{43}	a_{31}	a_{32}
0.18	0.24	0.58	0.27	0.73
$c_1 (s^{-1})$	$c_2 (s^{-1})$	$c_3 (s^{-1})$	$c_4 (s^{-1})$	
62000	50000	70000	5000	
$k_{31} (s^{-1})$	$k_{41} (s^{-1})$	$k_{51} (s^{-1})$	$P_{980} (s^{-1})$	
140000	185000	500000	280000	

7. Fig. 2(b): The authors need to provide a convincing explanation to the emission intensity decrease with increasing excitation intensity after reaching a threshold, correlating to their simulations. Their simulated results do not predict such remarkable intensity decrease (supplementary Fig. S1).

The emission intensity decreases after a plateau of maximum value reached is due to the high excitation power density used, which re-populate the intermediate excited states, e.g. 3H_4 (emit 800 nm emissions) to higher energy levels, e.g. 1D_2 (emit 455 nm emissions). Based on the parameters in table S3, we calculated UCNPs' power dependent curve on 455 nm and 800 nm as shown in the Figure attached below.

Since this region in the saturation curve is irrelevant to the resolution of NIRES nanoscopy, we think it is better not to emphasize this effect here.

Figure 1. The simulated saturation intensity curve of the 800 nm and 455 nm emissions from UCNPs.

8. Lines 107-123, pages 5-6: The authors used the “onset value” of the emission saturation curve to have discussion, which is too vague. The “onset value” should be quantified quantitatively. The resolution in ESSat, GSD-like, or FED microscopy is fundamentally related with the emission saturation properties of the probes. By modeling the relationship between the excitation intensity and emission intensity, it is very straightforward and thus kind of a routine to quantify the imaging resolution. The authors should perform resolution analysis more quantitatively, as done in previous papers.

We have already used theoretical simulation to achieve a quantitative resolution analysis in both Figure S2 and Figure 3. The simulation matches well with experimental result. This is the best we can possibly do here.

As shown in Figure S1, the saturation behaviour for rare earth doped materials is very different from that of dyes (2-photon), one parameter (I_s) used in normal STED and GSD is not enough to characterize and describe the saturation behaviour of upconversion nanoparticles. We calculated the resolution based on both I_s and the curvature of saturation curve.

We appreciate the reviewer for pointing out the definition of “onset value” was missing. **We added** “which is defined as the power point (in unit of I_s) to achieve e^{-2} of the maximum emission intensity” in our revised main text at line 108 in page 5.

9. In Figure 4, the authors claimed that they were doing single-particle imaging. Is there any evidence for that it is single particle?

We conclude they are single nanoparticles based on both their emission intensity and super-resolution image.

The standard deviation (SD) in emission intensity from the batch of single nanoparticles used in this work was less than 10%. We deduced the individual spots with emission intensity within SD range are single nanoparticles, as previously confirmed by TEM-confocal/wide field correlative microscopy methods^{5,6,7,8,9}. In our

case, it is impossible to conduct correlative microscopy of tissue covered sample under TEM. The best way is to use the intensity to tell. In each of depth inside tissue sample, we typically found ten emission spots. If the emission intensity from each of spots is within the standard deviation range of their average value, we can conclude that these individual spots were from single nanoparticles. To clarify this, we added “Benefiting from the controlled synthesis of intensity monodispersed UCNPs, each single nanoparticle can be distinguished from their clusters from clusters by comparing their emission intensity to statistical averaged value^{5,6,7,8,9}.” into Figure 4’s caption at line 157-159 in page 7.

Besides, the SD of particle size (from TEM measurement) from the batch of single nanoparticles used in this work was less than 5%. In that case, if there are two nanoparticles, our NIRES system can directly distinguish them.

10. The authors mentioned several times that their method is analogous to, “‘negative’ Ground State Depletion (GSD-like) mode” (line 59, page 2), “Similar to the principles of ESSat or ‘negative’ GSD” (line 97-98, page 4), and “Similar with GSD or ESSat sub-diffraction imaging approaches...”. The authors are suggested to avoid ambiguity and to describe their method more accurately by clarifying if their technique is GSD, ‘negative’ GSD or ‘negative’ GSD-like.

Our NIRES technique is unique in exploring the non-linear photonics properties of upconversion nanoparticles. To make this clear,

- (1) **We changed text between line 52 and 56 in page 2 to** “Here we report that setting the 980 nm excitation laser to create a doughnut beam enable super resolution imaging of single UCNPs through deep tissue. Low power coherent excitation at 980 nm can easily saturate the metastable level that emits bright NIR emission (800 nm), and the nonlinear power-dependent emission curve (saturation curve) is sharp. Both play the essential role in enabling a new mode of Near-Infrared Emission Saturation (NIRES) nanoscopy, ideal for deep tissue super-resolution imaging.”
- (2) **We changed text between line 92 and 95 in page 4 to** “To generate a super-resolution image of single nanoparticle by NIRES, a tightly-focused and doughnut-shaped excitation beam is used to scan across a sample containing UCNPs. Only when a single UCNP sits in the middle of the doughnut beam, NIRES generates a negative contrast. By using the definition of resolution in GSD microscopy¹⁰,”
- (3) **We changed text between line 103 and 104 in page 5 to** “The resolution of NIRES at a certain excitation power density is essentially determined by the emission saturation curve (Fig. 2b).”

We added text that “and long-term tracking of single molecules in deep¹¹.” at line 199-200 in page 9.

We changed text at line 22-24 in page 1 to “resolution of sub 50 nm, 1/20th of the excitation wavelength, in imaging of single UCNP through 93 μm thick liver tissue. This method offers a simple solution for deep tissue super resolution imaging and single molecule tracking.”

Reference

1. Urban, N. T., Willig, K. I., Hell, S. W. & Nägerl, U. V. STED nanoscopy of actin dynamics in synapses deep inside living brain slices. *Biophys. J.* **101**, 1277–1284 (2011).
2. Giannios, P. *et al.* Visible to near-infrared refractive properties of freshly-excised human-liver tissues: Marking hepatic malignancies. *Sci. Rep.* **6**, 1–10 (2016).
3. Giannios, P. *et al.* Complex refractive index of normal and malignant human colorectal tissue in the visible and near-infrared. *J. Biophotonics* **10**, 303–310 (2017).
4. Liu, Y. *et al.* Amplified stimulated emission in upconversion nanoparticles for super-resolution nanoscopy. *Nature* **543**, 229–233 (2017).
5. Ma, C. *et al.* Optimal Sensitizer Concentration in Single Upconversion Nanocrystals. *Nano Lett.* **17**, 2858–2864 (2017).
6. Zhao, J. *et al.* Single-nanocrystal sensitivity achieved by enhanced upconversion luminescence. *Nat. Nanotechnol.* **8**, 729–734 (2013).
7. Ma, C. *et al.* Probing the Interior Crystal Quality in the Development of More Efficient and Smaller Upconversion Nanoparticles. *J. Phys. Chem. Lett.* **7**, 3252–3258 (2016).
8. Wang, F. *et al.* Microscopic inspection and tracking of single upconversion nanoparticles in living cells. *Light Sci. Appl.* **7**, 18006–18007 (2018).
9. Yang, Y., Zhu, Y., Zhou, J., Wang, F. & Qiu, J. Integrated Strategy for High Luminescence Intensity of Upconversion Nanocrystals. *ACS Photonics* **4**, 1930–1936 (2017).
10. Rittweger, E., Wildanger, D. & Hell, S. W. Far-field fluorescence nanoscopy of diamond color centers by ground state depletion. *EPL* **86**, 14001 (2009).
11. Jin, D., Xi, P., Wang, B., Zhang, L. & Oijen, A. M. Van. Nanoparticles for super-resolution microscopy and single-molecule tracking. *Nat. Methods* **15**, 415–423 (2018).